# The effectiveness of global protected areas for climate change mitigation

L. Duncanson [1] ✉, M. Liang[1], V. Leitold[1], J. Armston[1], S. M. Krishna Moorthy[1], R. Dubayah [1], S. Costedoat[2], B. J. Enquist[3,4], L. Fatoyinbo [5], S. J. Goetz [6], M. Gonzalez-Roglich[7], C. Merow [8], P. R. Roehrdanz [2], K. Tabor[5,9] & A. Zvoleff[2]

Forests play a critical role in stabilizing Earth's climate. Establishing protected areas (PAs) represents one approach to forest conservation, but PAs were rarely created to mitigate climate change. The global impact of PAs on the carbon cycle has not previously been quantified due to a lack of accurate global-scale carbon stock maps. Here we used ~412 million lidar samples from NASA's GEDI mission to estimate a total PA aboveground carbon (C) stock of 61.43 Gt (+/− 0.31), 26% of all mapped terrestrial woody C. Of this total, 9.65 + /− 0.88 Gt of additional carbon was attributed to PA status. These higher C stocks are primarily from avoided emissions from deforestation and degradation in PAs compared to unprotected forests. This total is roughly equivalent to one year of annual global fossil fuel emissions. These results underscore the importance of conservation of high biomass forests for avoiding carbon emissions and preserving future sequestration.

Earth's ecosystems play a critical role in the carbon cycle, with estimates of global terrestrial aboveground carbon (AGC) of ~308 Gt in 2010[1,2] and annual uptake of ~8 Gt $CO_2$[3]. The primary causes of AGC loss are deforestation and forest degradation, while vegetation carbon sinks are associated with afforestation and forest recovery. Several policy frameworks emphasize that habitat conservation and restoration should contribute simultaneously to biodiversity conservation and climate change mitigation[4]. These frameworks include the UN Sustainable Development Goals (SDGs), decisions under the United Nations Framework Convention on Climate Change (UNFCCC) and the Convention on Biological Diversity (CBD). To support goal setting and the implementation of international strategies and action plans, guidance is needed to identify how well-protected areas contribute to maximizing synergies between conserving biodiversity and other ecosystem services such as climate change mitigation[5].

Forest conservation is a crucial mechanism for forest management toward climate change mitigation, and for curbing biodiversity loss[6,7]. Protected areas are a foundation for global forest conservation efforts and monitoring PA effectiveness is key for determining progress in achieving the UN SDGs[8]. While most efforts to establish PAs have been focused on biodiversity protection[9], there are clear co-benefits of biodiversity and carbon conservation efforts, as older, biodiverse forests also typically store more carbon[5,6,10]. PAs have been demonstrated to effectively avoid forest cover loss in many regions[11–13], as well as regulate temperature and local climate[14], and potentially boost carbon sequestration capacity[15,16]. Therefore, PA expansion may be a pathway to bolster climate change mitigation[17]. Intact forests, especially tropical forests, can sequester twice as much carbon than more human-impacted forests and planted monocultures[16,18]. Protected forest areas are thought to contribute a large fraction (~27%) of the net global GHG sink[3] but large uncertainties remain in the magnitudes of AGC stocks

[1]Department of Geographical Sciences, University of Maryland, College Park, MD, USA. [2]Moore Center for Science, Conservation International, Arlington, VA 22202, USA. [3]Department of Ecology and Evolutionary Biology, University of Arizona, Tucson, AZ 85721, USA. [4]The Santa Fe Institute, 1399 Hyde Park Road, Santa Fe, NM 87501, USA. [5]NASA Goddard Space Flight Center, Greenbelt, MD, USA. [6]School of Informatics, Computing and Cyber Systems, Northern Arizona University, Flagstaff, AZ, USA. [7]WCS, General Roca, Río Negro, Argentina. [8]Eversource Energy Center and Department of Ecology and Evolutionary Biology, University of Connecticut, Storrs, CT, USA. [9]Department of Geography and Environmental Systems, University of Maryland Baltimore County, Baltimore, MD, USA. ✉e-mail: lduncans@umd.edu

and fluxes in terrestrial ecosystems[19,20]. As a result, the degree to which protected status contributes to avoided carbon emissions or enhanced sequestration at a global scale remains highly uncertain.

Here, we analyze millions of spaceborne lidar-derived estimates of AGC from NASA's Global Ecosystem Dynamics Investigation (GEDI)[21] to spatially quantify the carbon effectiveness of PAs and test the assumption that these areas provide disproportionately more ecosystem services through carbon storage and sequestration than non-protected areas[22]. Previous attempts to quantify carbon content in PAs had high uncertainties and/or biases, as past satellite biomass products are known to saturate in high biomass forests[23], such as old-growth PAs. GEDI is the first satellite lidar system designed specifically to map forest structure, and provides orders of magnitude more 3D samples of Earth's forests than have previously been available, capable of collecting accurate data in even the densest and tallest forests[23]. GEDI launched on December 5, 2018, and is collecting full-waveform lidar samples from the International Space Station (ISS) between ~52°N and 52°S under the ISS orbit (Fig. 1). GEDI has three lasers operating at 1064 nm, each illuminated ~25 m circular "footprints" (circular pixels) to produce billions of high-resolution samples of surface elevation, vegetation height, and foliage distribution. GEDI is not a mapping mission, in that it does not collect data continuously over Earth's surface, but instead provides samples spaced ~60 m apart along each laser track, with ~600-m spacing between tracks. Therefore not every area of every PA is mapped. 25 m samples are aggregated to 1 km estimates of Aboveground Biomass Density (AGBD, which is subsequently converted to AGCD)[24]. At the time of writing, GEDI has collected sufficient data to fill ~70% of all GEDI-domain 1 km pixels. GEDI collects data from the International Space Station (ISS), which covers all tropical and temperate forests, as well as the southern boreal, but does not collect data north of ~52° latitude. Thus, while the results in this paper are global scale, they are not truly global as they omit PAs north of ~52°.

GEDI provides a richer dataset to quantitatively address questions of forest C stocks and fluxes than have been previously available. We use GEDI's data to quantify the additional AGC stocks attributed to the existence of PAs (termed "carbon effectiveness" of PAs) at a global scale (within the GEDI domain). This is achieved through matching each PA to ecologically similar unprotected areas, or counterfactuals (based on climate, human pressure, land type, country, and other factors). Our analysis is based on the richness of GEDI samples within globally distributed terrestrial PAs, which allows us to assess the enhanced value of C stocks in PAs relative to their unprotected counterfactuals. In the GEDI domain, ~26% of aboveground C falls within PAs, where avoided emissions from deforestation and degradation are lower than in unprotected counterfactuals. A global estimated 9.65 Gt of avoided emissions are attributed to PA status. The largest carbon effectiveness is found in tropical moist forests, specifically in Brazil, although every continent exhibits higher aboveground C stocks in PAs. These results underscore the importance of PAs in the global carbon cycle and for climate change mitigation.

## Results

We estimate the GEDI-domain (N/S of 52° latitude) total protected area woody AGC is 61.4 Gt (+/− 0.31 Gt). While PAs represent ~11% of the measured forested area (16.2 Mkm², Fig. 1), they store 26% (61.4 Gt) of the total estimated AGC (235 Gt)[24]. This represents all aboveground woody carbon stocks in PAs, not just those attributed to PA status (termed additionally preserved AGC). Areas with PA status have, on average, 28% more AGC than their matched unprotected sites, for a total of 9.65 Gt (+/− 0.9 Gt) of additionally preserved PA AGC.

### Why is there more biomass in protected areas than in similar forests?

Most forested PAs (62.7%) had significantly higher AGC in 2020 than matched unprotected areas. PAs were matched to unprotected areas

using a suite of ecological, anthropogenic pressure, and climate variables representing conditions in 2000 (Supplementary Table S1). We therefore assume AGC densities in our samples within PAs and counterfactuals were equal at that time. By then comparing 2020 GEDI measurements between these protected/unprotected pairs, differences in 2020 represent approximately 20 years of change associated with PA status. The observed differences in 2020 structure could be explained by (i) less AGC loss in PAs compared to unprotected counterfactuals resulting from deforestation and/or forest degradation between 2000 and 2020, (ii) increased forest growth in PAs compared to counterfactuals between 2000 and 2020, or (iii) PAs being preferentially established in higher biomass (C) areas before 2000.

Forest cover dynamics from the Landsat data record were analyzed from 2000 to 2020[25], and show more than half of PAs with >2.5 Mg/ha higher mean AGC also had lower rates of forest loss within PAs than in unprotected counterfactuals (Supplementary Fig. S3). Thus, the observed higher concentrations of AGC in PAs are attributed primarily to avoided carbon emissions from deforestation, which is supported by the optical data record (hypothesis i). In ~18% of PAs, the forest cover change data in counterfactuals did not detect loss, while GEDI still observes higher AGC in PAs. In these cases, we speculate that degradation is occurring outside of PAs, but is not visible to passive optical sensors that form the basis of the forest cover loss data (e.g., small-scale logging, understory loss, etc.). This apparent degradation signal demonstrates the importance of datasets such as GEDI to detect subtle changes in carbon stocks that are not detectable with previous satellite datasets[26]. Indeed, avoided degradation associated with PAs has likely been missed in past studies analyzing reduced carbon loss rates in PAs, and thus underestimated. Although we attribute PAs where we see higher AGC without reduced forest cover losses as avoided degradation, PA vegetation in these cases may also be exhibiting enhanced regrowth compared to unprotected forests (ii)[22,27]. Our assertion of enhanced regrowth is supported by local and regional studies assessing PA forest growth[28,29]. Based on these results, we cannot definitively attribute the signal in this 18% of PAs to avoided degradation, enhanced growth, or a combination of both. Regardless, there is a clear signal of higher C densities here, and attribution of higher C in these PAs would benefit from further investigation with conservation practitioners, including fieldwork. Overall, we therefore attribute our findings primarily to avoided emissions from deforestation and secondarily to a combination of enhanced growth and/or avoided degradation (hypotheses i and ii).

We found little evidence that PAs were placed in higher AGC density forests (hypothesis iii). If PAs were being established in higher AGC areas, their baseline (year 2000) AGC should be higher than counterfactuals. We used a pre-existing year 2000 AGC map to test this[3], and found that recently established PAs (established in or after 2000) had little differences in AGC between PA and matched unprotected area in the year 2000 (Supplementary Fig. S4). Older PAs did have significantly higher 2000 AGB values than matched areas, and this difference increased with time since establishment. These findings are in line with expectations of PAs adding additional AGC through time, rather than being preferentially located in carbon-dense locations. We therefore conclude that preferential establishment in high AGC areas does not explain our observations of higher AGC densities in PAs in 2020.

### The Amazon dominated the global signal

The starkest contrast between protected and unprotected counterfactuals was found in South America, specifically the Tropical and subtropical moist broadleaf forest biome in the Brazilian Amazon (Fig. 2). This supports recent results related to the effectiveness of PAs in Brazil for avoiding deforestation[30,31], and quantifies the climate change impact of Brazilian PAs at 3.54 Gt AGC more than unprotected counterfactuals, representing 36.6% of the global signal. Again, this

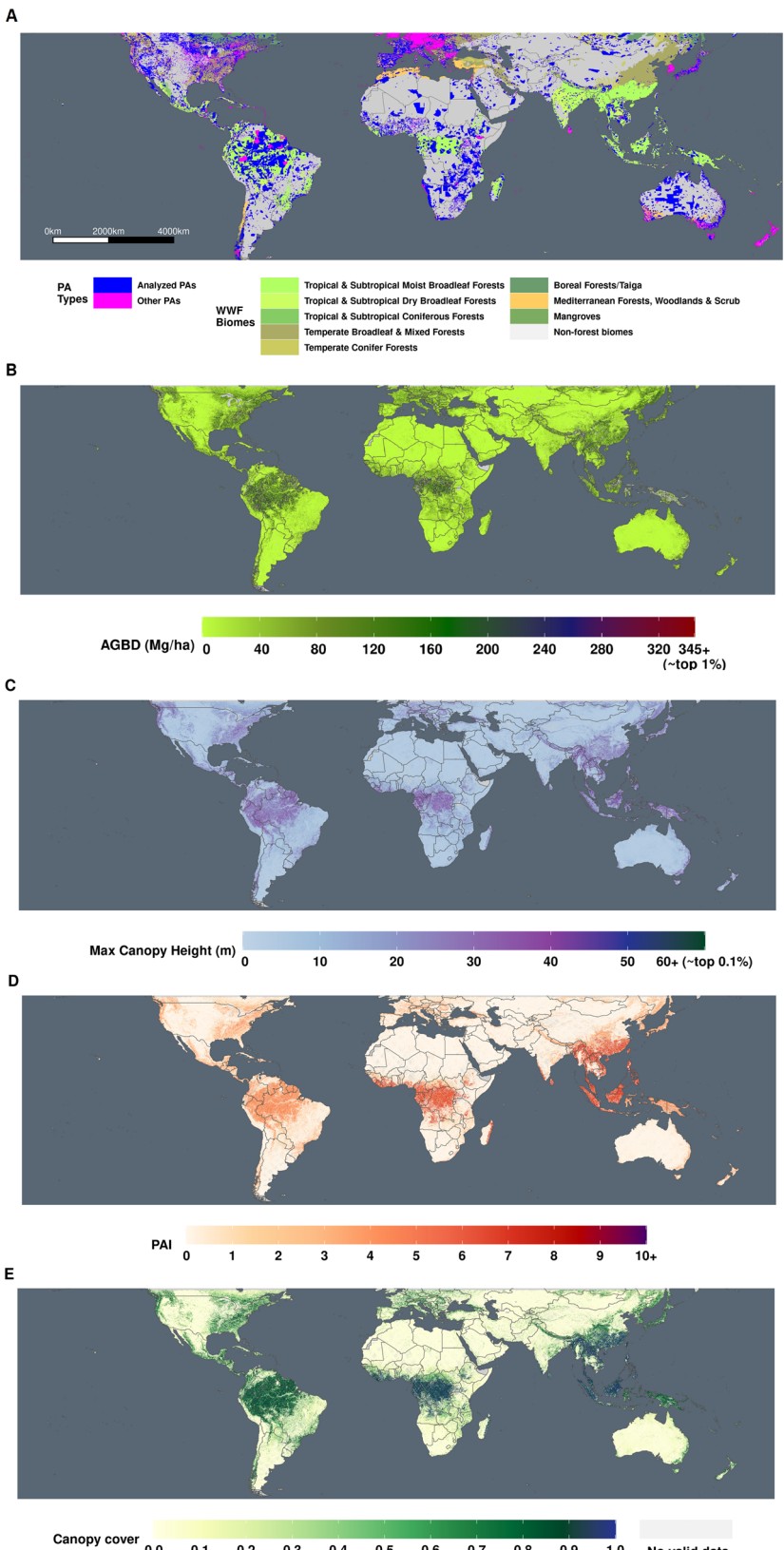

**Fig. 1 | Global-scale vegetation 3D structure data from NASA's GEDI mission.** The GEDI-domain PAs cover a range of biomes (**A**). GEDI AGBD (**B**), canopy cover (**C**), canopy height (**D**), and Plant Area Index (PAI, **E**) were analyzed for all PAs and unprotected counterfactuals to establish the forest structure implications of PAs. World base map made with Natural Earth.

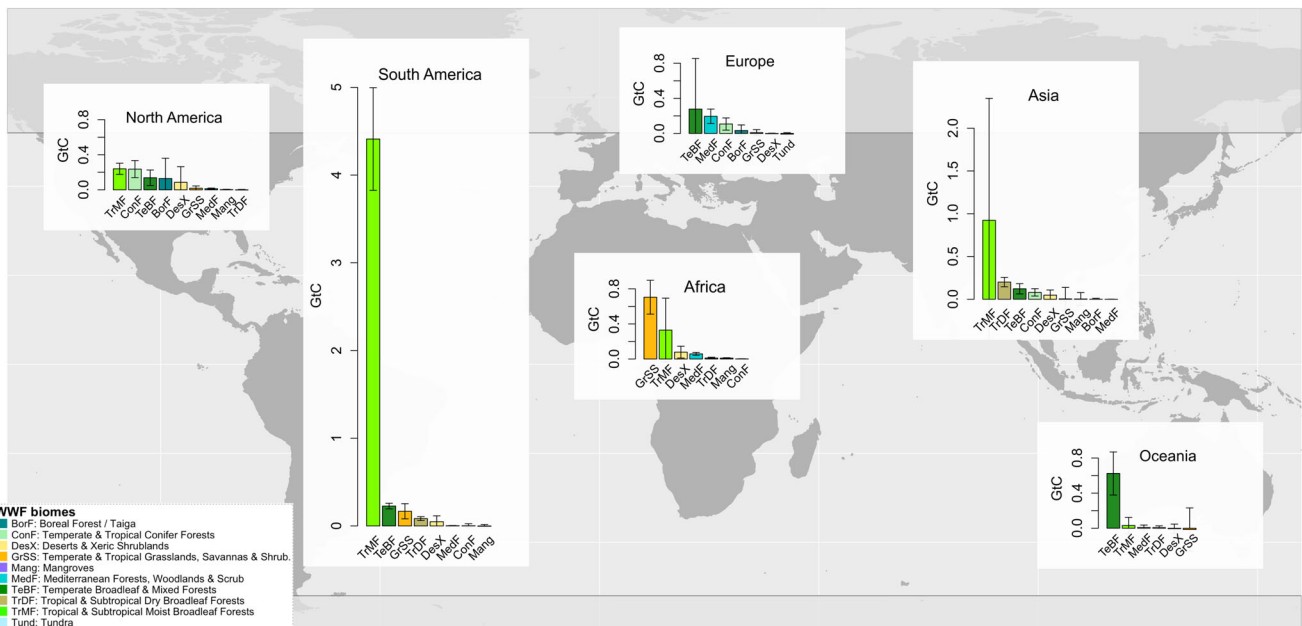

**Fig. 2 | Total additionally preserved AGC aggregated by continent and biome.** PAs effectively preserve additional AGC across continents and biomes, with forest biomes dominating the global signal, particularly in South America. The additional preserved AGC (Gt) in WWF biome classes (total Gt +/− SEM*area). World base map made with Natural Earth. The full set of analyzed GEDI data are represented in this figure (n = 412,100,767).

supports our hypothesis that differences between PA and unprotected AGC are primarily associated with avoided emissions from forest loss and degradation, as Brazil experienced the highest national forest loss rates of any country during the analysis period[12]. It is also noteworthy that while South American PAs have the greatest avoided carbon emissions (additionally preserved AGC), they cover roughly the same geographic extent as African PAs (Table 1). Avoided emissions are a factor of both (a) deforestation rates outside of PAs and (b) the C densities of forests being lost. Therefore while Africa hosts a similar total area of PAs, these have, on average, lower C densities than in Latin America or tropical Asia[32], and are smaller (the median PA area in Africa is 85 km$^2$ compared to South America's 127 km$^2$), which may result in increased disturbance. In addition, there is a larger increase in anthropogenic pressure in both PAs and counterfactuals in the Afro-tropics than in South America, which may be reflected in the relatively lower carbon effectiveness we saw in African forests[33]. Indeed, Africa has the largest proportion of PAs with no additionally preserved AGC (Supplementary Fig. S5).

PA additionally preserved AGC varied considerably by biome, and while the signal was unsurprisingly highest in tropical moist forests, the dominant biome varied by continent (Fig. 2). The Amazon dominates the global signal in carbon effectiveness. Tropical moist forests also dominate Asia's signal, given high forest loss rates in Southeast Asia. Conversely, African PA effectiveness was dominated by temperate and tropical/subtropical grasslands, savannas, and shrublands. Indeed, Africa is the only continent where the PA effectiveness is not highest in a forest-dominated biome, suggesting that PAs in woodlands, grasslands, savannas and shrublands may be reducing land conversion, e.g., to agriculture, reducing charcoal degradation[34], or bolstering woody encroachment[35,36] and thus curbing net carbon emissions in these systems. Yet tropical dry forests and woodlands are under less protection than forests both in Africa (less than one-fourth protected) and worldwide (less than one-third protected)[37]. With an estimated population of 320 million inhabiting such landscapes in the 2000s and an average of 2.4% increase per annum in sub-Saharan Africa[38,39], these ecosystems are facing higher human development pressure than humid forests. Therefore, our results substantiate

the critical roles of protected areas in dry forest and woodland ecosystems.

At a global scale, forests dominate the carbon effectiveness of PAs (Table 1 and Fig. 2). A singular exception is mangrove forests, which show a near zero effect of PA on AGC stocks. This may be due to a few factors. First, many mangroves globally are below 5 m in height, therefore, GEDI may miss a large portion of mangrove biomass both in protected areas and outside of them, which likely limits our analysis. Second, mangrove PAs may either be ineffective at protecting AGC as we know that mangroves are extremely vulnerable to human pressure. Specifically, we found lower AGC in protected mangrove areas in Indonesia and Malaysia, which also harbor most of the mangrove cover in mangrove PAs worldwide (25% of global mangrove extent and C is in Indonesia alone, as well as most of the deforestation). Finally, deforestation rates have declined in all mangroves since the year 2000[40] and 50% or more of global mangrove cover was already lost by 2000, limiting remaining unprotected mangrove areas available for cutting. Our mangrove results contrast with studies demonstrating effective PAs for curbing mangrove loss in Indonesia[41]. However, our results may be related to complicated and challenging mangrove management[42], and are supported by results of a global mangrove study[40] which indicated increased pressure on Indonesian-protected mangroves in particular, providing evidence that some PAs may be ineffective at protecting mangrove AGC. It is possible that mangrove AGC is being degraded while canopy cover remains intact, but further research specifically into C-rich mangrove ecosystems in PAs is critical.

Most countries (78%) in the GEDI domain have higher AGC stocks in PAs compared to counterfactuals. The top 20 countries in terms of PA effectiveness at preserving carbon (Fig. 3) are either (a) geographically large, (b) host forests with high AGC, and/or (c) have high forest loss rates of unprotected forests (Fig. 4). Many of the top 20 countries fall in tropical dense forested areas such as the Amazon (Brazil, Venezuela, Peru, Bolivia), the Congo Basin (DRC), or Southeast Asia (Thailand, Indonesia, Cambodia, Malaysia). Outside the tropics, highly ranked countries tend to be geographically large (Australia, USA, Chile, France, Spain), or clustered in Eastern or Southern Africa,

**Table 1 | The additionally preserved AGC is the difference between the AGC observed in PAs and ecologically similar unprotected counterfactuals**

| | Additional PA C (Gt) | Total above-ground C in PAs (Gt) | Total PA area (Mkm²) |
|---|---|---|---|
| **Globe** | | | |
| Globe | 9.64 +/−0.9 | 61.41 +/− 0.3 | 16.15 |
| **Continent** | | | |
| South America (SA) | 4.94 +/−0.47 | 29.12 +/− 0.16 | 4.49 |
| Asia (AS) | 1.38 +/−0.59 | 9.08 +/− 0.21 | 2.05 |
| Africa (AF) | 1.2 +/−0.17 | 9.58 +/− 0.07 | 4.33 |
| North America (NA) | 0.86 +/−0.33 | 5.99 +/− 0.1 | 2.32 |
| Oceania (OC) | 0.65 +/−0.32 | 2.26 +/− 0.09 | 1.65 |
| Europe (EU) | 0.61 +/−0.13 | 5.37 +/− 0.05 | 1.32 |
| **Biome** | | | |
| Tropical and subtropical moist broadleaf forests | 5.93 +/−1.59 | 37.53 +/− 0.48 | 4.46 |
| Temperate broadleaf and mixed forests | 1.39 +/−0.64 | 7.8 +/− 0.22 | 1.39 |
| Tropical and subtropical grasslands, savannas and shrublands | 0.81 +/−0.31 | 5.25 +/− 0.11 | 3.17 |
| Temperate Coniferous Forest | 0.4 +/−0.13 | 3.13 +/− 0.07 | 0.55 |
| Tropical and subtropical dry broadleaf forests | 0.3 +/−0.06 | 1.13 +/− 0.02 | 0.32 |
| Mediterranean Forests, woodlands and scrubs | 0.28 +/−0.09 | 0.93 +/− 0.06 | 0.55 |
| Deserts and xeric shrublands | 0.25 +/−0.22 | 1.06 +/− 0.09 | 2.88 |
| Boreal forests/Taiga | 0.16 +/−0.24 | 1.73 +/− 0.03 | 0.74 |
| Temperate grasslands, savannas and shrublands | 0.05 +/−0.08 | 0.26 +/− 0.02 | 0.36 |
| Flooded grasslands and savannas | 0.04 +/−0.03 | 0.22 +/− 0.02 | 0.28 |
| Tropical and subtropical coniferous forests | 0.03 +/−0.03 | 0.51 +/− 0.01 | 0.10 |
| Mangroves | 0.01 +/−0.08 | 0.22 +/− 0.03 | 0.09 |
| Tundra | −0.01 +/−0.02 | 0.09 +/− 0.01 | 0.74 |
| Montane grasslands and shrublands | −0.01 +/−0.15 | 1.19 +/− 0.04 | 0.55 |

This AGC is aggregated at a biome, continental, and global scale. The total PA AGC stock and total area of PAs in million km² (Mkm²) are also reported. Note GEDI's biomass (C) products only account for aboveground woody C, even in non-forest ecosystems (i.e., trees and shrubs, not herbaceous C or soil C stocks). In addition, North America and Europe are underestimated as PAs north of ~52° latitude were not included in this study.

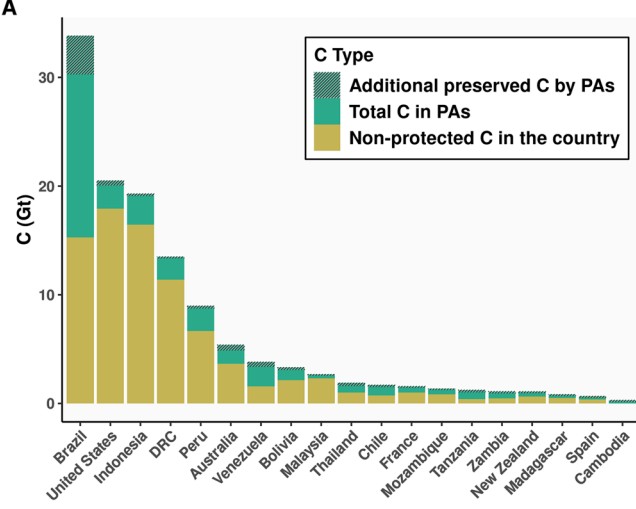

**A**

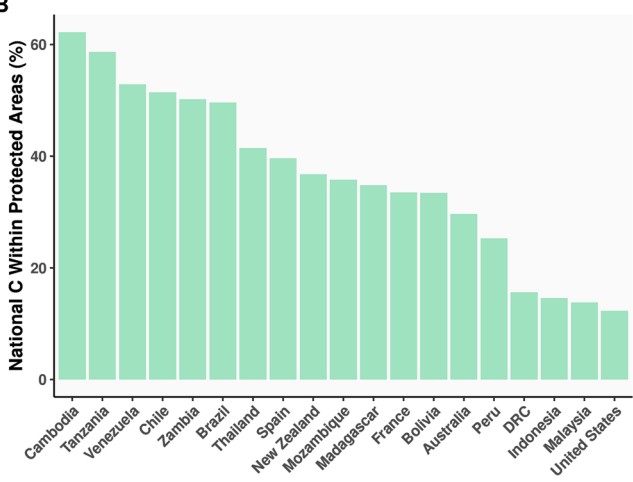

**B**

*Countries completed covered by GEDI data are included; US does not include Alaska*

**Fig. 3 | In the top 20 countries with the most carbon-effective PAs, most AGC remain unprotected.** The top 20 countries in total PA additionally preserved AGC, ranked by total national AGC (**A**) and the proportion of national AGC stored in PAs (**B**). More than half of the national AGC have protected status for six countries, while most AGC does not have protected status for the remaining 14.

hydrology[45] and be tightly linked to climate and soil characteristics[46]. These results therefore highlight co-benefits of PAs between carbon and biodiversity[10].

## Discussion
Our results highlight the critical importance of protected areas to help mitigate climate change. Aboveground carbon flux is only one way forests influence climate change, while forest loss also influences albedo, evaporative flux, belowground biomass etc., which are also likely impacted by protected status[22]. This study focused specifically on the avoided C emissions associated with preservation of woody aboveground C stocks, and thus likely underestimate the full climate impact of PA status. While our findings support results from past studies which analyze the effectiveness of PAs for preventing forest loss[10,28,47,48], our results extend past studies using next-generation satellite data to directly quantify enhanced stocks and/or avoided carbon emissions from deforestation and degradation with great consistency and accuracy. The majority of PAs exhibited avoided emissions compared to unprotected counterfactuals. In some areas, we see a relatively small difference between PA and matched AGB

where our results show biggest impacts outside of forests (Tanzania, Mozambique, Zambia).

**PAs are characterized by taller, denser, higher biomass forests**
While we focused primarily on an analysis of forest AGC, similar trends were found for other GEDI-based forest structure variables, including maximum and mean canopy height, Plant Area Index (PAI), and canopy cover (Fig. 5). As GEDI predicts AGBD (which we convert to AGCD) as a function of height metrics[43], similar effectiveness was anticipated between AGC and height. However, canopy cover and PAI, which are independent structural data products, were also higher within PAs than outside PAs. This suggests forest structure beyond carbon is being negatively affected in the absence of protected status, which correlates with habitat suitability and biodiversity[44]. Further, forest structure (e.g., complexity, cover) is known to influence regional

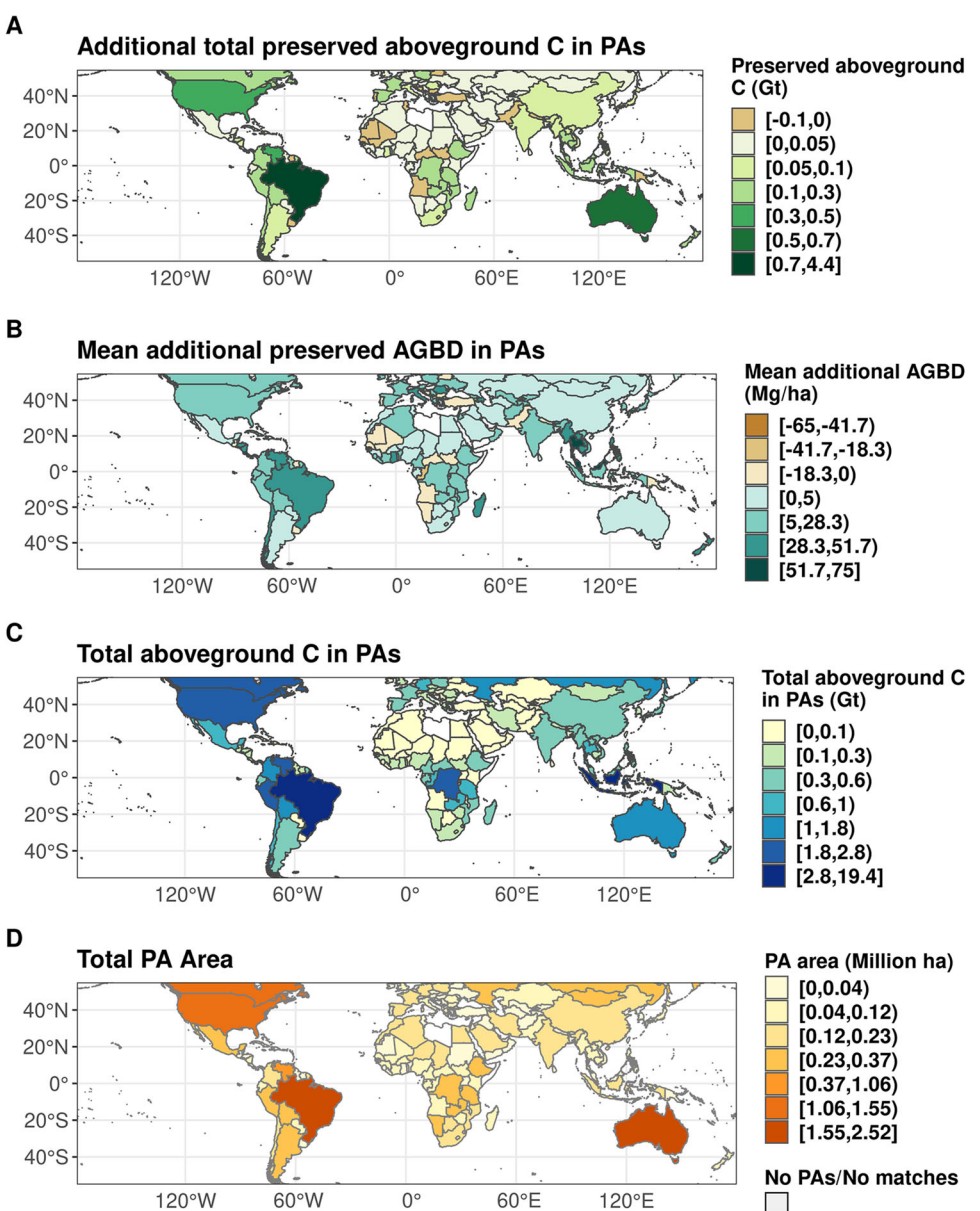

**Fig. 4 | PA carbon effectiveness by country.** Nationally aggregated additionally preserved AGC (**A**) is a function of the national total stored GEDI AGBD in PAs (**B**), the average C effectiveness of PAs in a country (difference between PA AGC and matched unprotected AGC, **C**), and the total PA area per country (**D**). World base map made with Natural Earth.

levels, which suggests either reduced effectiveness of PAs, or lower pressure on forests regardless of PA status. In these countries, our data cannot determine whether PAs are ineffective or anthropogenic pressure is low, and results should be interpreted in the context of national data. The correspondence between high effectiveness of PAs for AGC preservation in regions with high forest loss rates (e.g., Brazil[49], Southeast Asia[50]) clearly highlights that maintaining and expanding PAs, particularly in high C forests, is essential for achieving global carbon mitigation goals.

Our analysis used a satellite lidar dataset to quantify the carbon effectiveness of protected areas, which has not previously been possible with the accuracies or spatial resolution enabled by the GEDI mission. Further, GEDI is sensitive to structural change in forests that do not cause full canopy cover removal, and thus past studies likely have not accounted for avoided degradation in PAs. Time series satellite records[3] indicate that these findings are related to avoided emissions from deforestation, but several important caveats should be

noted. First, we do not account for leakage (i.e., deforestation pressures that move away from protected areas), and it is possible that background deforestation rates would be lower if not enforced in protected areas[51]. However, evidence shows that leakage is unlikely to negate our findings[52,53]. Secondly, we do not analyze which type of PA is the most effective, analyze PAs by governance type, or subdivide PAs into multiple classes, but instead focus on a necessarily simplified global analysis. Results from Tanzania suggest that multiple designations of PAs bolster carbon effectiveness, but that analysis of the impact of PA designation likely varies by nation[54]. GEDI data might be crucial for Indigenous Territories, which may be at higher risk for degradation[55], but also may be more resilient to deforestation pressure[56].

Our analysis focused on aboveground woody carbon but does not account for belowground carbon (BGC) or the future added sequestration from maintaining higher AGC within PAs[22]. Therefore, our AGC totals only account for part of PAs' full current and future effectiveness

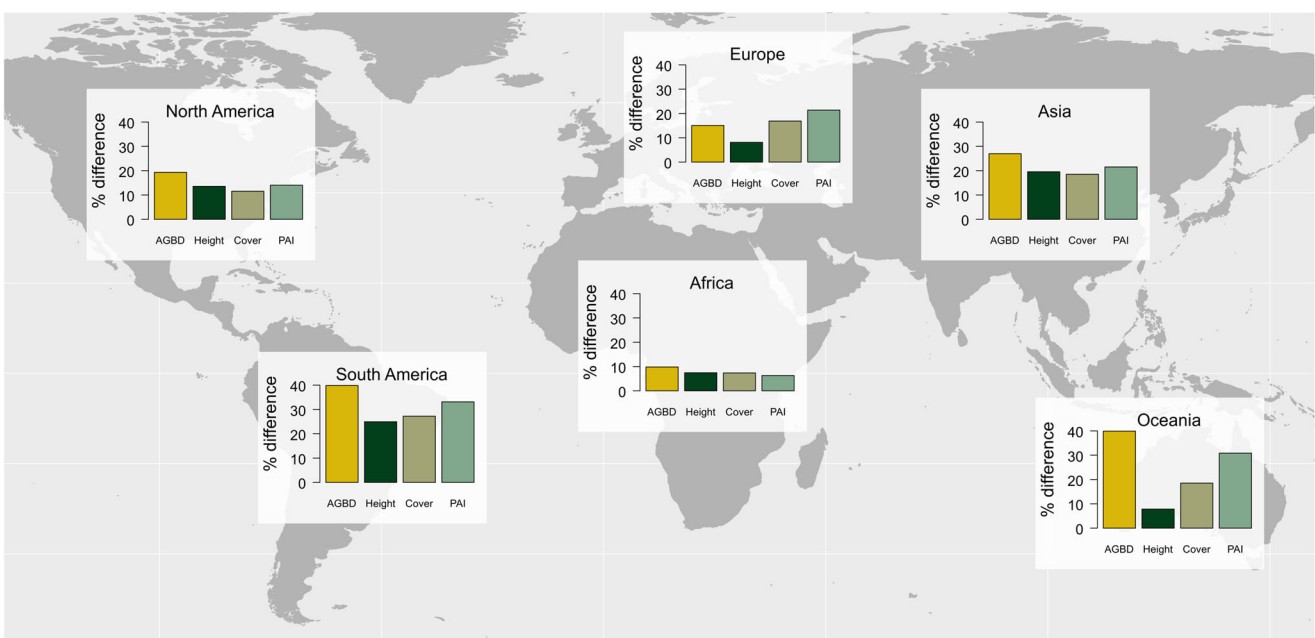

**Fig. 5 | Differences in AGBD, forest height, canopy cover, and PAI between PAs and matched unprotected areas.** PA status resulted in increased forest height and density, showing generally similar trends (higher values within the PA) across structural metrics and continents. World base map made with Natural Earth.

for AGC preservation. In addition, we do not account for the future protection and enhancement of carbon sequestration potential related to leaving PA AGC intact. Our results summarize the aggregate of avoided emissions and forest growth over the past two decades (post-2000), but these protected forests will continue to absorb carbon in the future, and thus we have made conservative estimates of their importance for avoiding carbon emissions, provided they remain intact[57].

Protected areas are effective for preventing carbon emissions related to deforestation and degradation, and as they store a substantial proportion of Earth's forest aboveground carbon, preservation of these regions is an essential Natural Climate Solution[7]. These areas are particularly critical in regions of the world experiencing high rates of deforestation. We also demonstrate that other forest structure metrics related to habitat (forest height, canopy cover, and PAI) are also preserved by PAs, suggesting effective co-benefits of PAs include both climate and biodiversity. Our work demonstrates that the protected area targets highlighted by the United Nations Convention on Biological Diversity will benefit the UNFCCC goals such as the Glasgow COP26 Leaders' Declaration on Forests and Land Use commitment to end deforestation by 2030. We provide a quantitative and globally consistent demonstration of the success of PAs to mitigate climate change and reiterate the multiple benefits of expanding these areas.

## Methods
We used NASA's Global Ecosystem Dynamics Investigation (GEDI) data to quantify the effectiveness of PAs at preserving forest structure and aboveground carbon. We used statistical matching to link PA 1-km pixels to pixels outside of PAs with similarity in terms of their ecology, environment, and human pressure using data layers from the year 2000[54]. These control pixels, or matched unprotected counterfactuals, allowed us to compare 2020 conditions between areas that would theoretically have the same biomass distributions in 2000 except for the presence of forest conservation through PA designation. We extracted distributions of GEDI forest structure measurements within PAs and matched counterfactuals, and quantified the differences between the two. Figure 6 provides a visual representation of the methods applied in this paper.

### Global ecosystem dynamics investigation mission overview and datasets
NASA's Global Ecosystem Dynamics Investigation (GEDI) is a lidar mission aboard the International Space Station (ISS) which has been collecting high-resolution (25 m) samples of 3D vegetation structure since 2019. GEDI produces a suite of forest structure products both at the 25 m sample resolution, and a gridded 1 km resolution. GEDI's lidar waveforms are processed to generate estimates of vegetation height metrics, terms "Relative Height" (RH) metrics representing the vertical distribution of foliage in each 25 m footprint (the area illuminated by each laser beam sample)[58]. RH100 represents the maximum canopy height per sample, while RH50 roughly represents mean height. Waveforms are also processed to estimate %canopy cover, and Plant Area Index (PAI), which is roughly the same as Leaf Area Index (LAI) but includes woody material[59]. Finally, Aboveground Biomass Density (AGBD) is estimated for each footprint by fitting models between thousands of globally distributed sets of waveform metrics and field plot measurements[43]. These models are parametric Ordinary Least Squares models that predict AGBD as a function of RH metrics. They are divided by continent and Plant Functional Type (PFT), and are broadly consistent, typically predicting AGBD as a function of RH98 (maximum height) and a lower RH metric, representative of roughly mean canopy height. The height, cover, PAI and AGBD products used in this study are from the first 18 months of mission data, between April 2019 and September 2020, and represent a total of >400 million 3D structure samples. The AGBD estimates are translated into AGCD by multiplying by 0.49 (the global average conversion from dry woody AGBD to AGCD). GEDI presents the first-ever global-scale satellite dataset designed specifically for measuring forest structure, and overcomes limitations for forest height, cover, and AGB mapping in dense forests where previous maps have little sensitivity to AGB[19].

### International union for conservation of nature datasets
The World Database of Protected Areas (WDPA) is a joint project between the United Nations Environment Programme (UNEP) and the International Union for Conservation of Nature (IUCN), and represents the most comprehensive global database of terrestrial and marine protected areas. The database is compiled and managed by the World

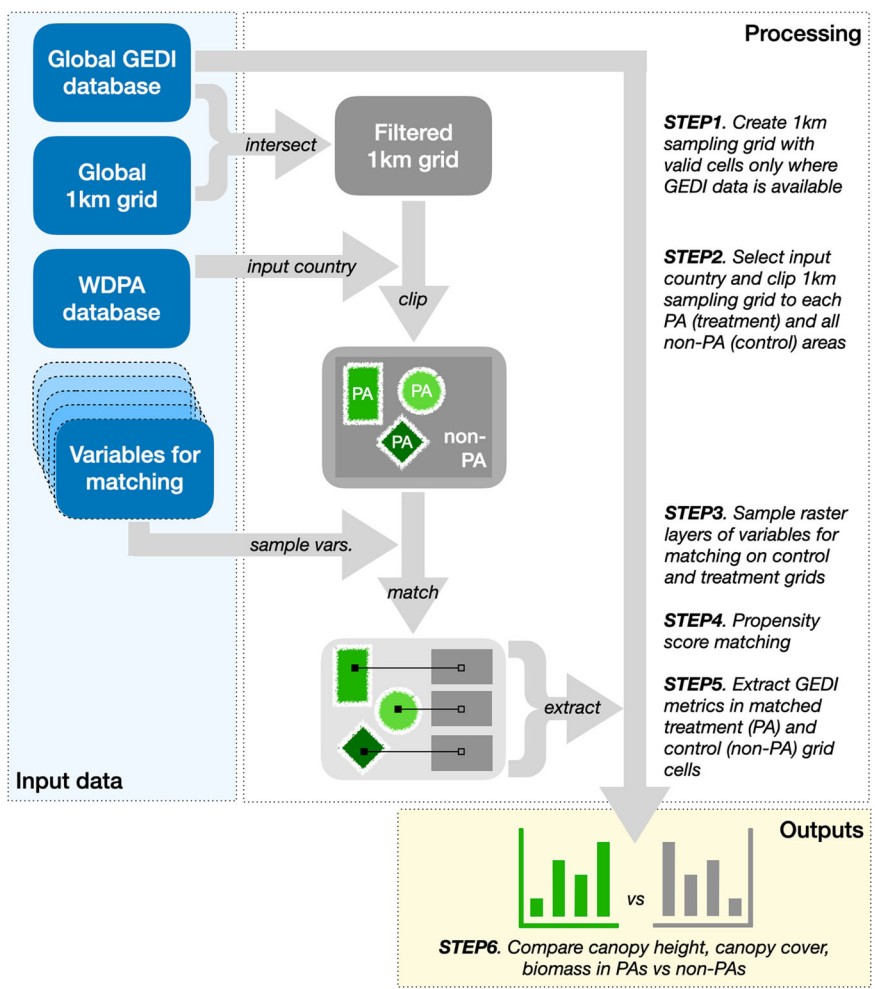

**Fig. 6 | Visualization of study methodology.** Protected area effectiveness is quantified by comparing distributions of GEDI vegetation structure measurements within PAs and matched control pixels.

Conservation Monitoring Centre of UNEP (UNEP-WCMC), in collaboration with governments, NGOs, academia, and industry. Each protected area in the WDPA must meet the IUCN definition of a clearly defined geographical space, recognized, dedicated and managed, through legal or other effective means, to achieve the long-term conservation of nature with associated ecosystem services and cultural values[60] and the Protected Planet data standards (see WDPA User Manual 1.6). IUCN classifies protected areas based on management category (I–VI, with increasing human intervention), governance type (i.e., who holds authority and accountability), status year, designation, and various other attributes. The WDPA is available online through ProtectedPlanet.net and is updated monthly. In this study, we used the September 2020 version of the WDPA, with a total of 262,804 protected areas (240,713 polygons and 22,091 points) covering 245 countries and territories (Fig. 7). We only used PAs from terrestrial, vegetated ecosystems within GEDI's area of coverage. The WWF biomes corresponding to the PAs analyzed in this study are shown in Fig. 2 and Table 1. In the accounting of total AGB in protected areas, overlapping protected areas are only counted once.

### Matching algorithm

To determine the effectiveness of PAs, we matched 1 km sample pixels within PAs to 1 km sample pixels outside of any PAs by controlling for a series of geophysical and socioeconomic characteristics (Supplementary Table S1). The choice of 1-km resolution for analysis is due to limitations in the spatial resolution of matching covariates and

computational resources. In addition, this resolution aligns with stakeholder and policy maker needs and with scientific products, such as the GEDI L4B 1-km gridded aboveground biomass[24]. A 10 km buffer around any PA borders was also implemented to avoid mixed pixels and remove potential spillover effects e.g., encroachment at the PA boundaries or potential leakage into neighboring forests[61–63]. Therefore no locations within the 10 km buffer of any protected area were considered for matching. Only 1-km PA pixels entirely within PA boundaries were considered. By comparing the vegetation structure in PAs and their matched unprotected counterfactuals, we addressed the fact that PAs were usually non-randomly distributed and assessed the conservation strategy's efficacy in storing biomass and carbon. To assign counterfactual 1 km pixels to PA 1 km pixels, we required exact matching of several covariates: land cover category, country, ecoregion and biome. In addition, we used propensity score matching for quantitative covariates: geophysical, climatic and social variables, in elevation, slope, mean precipitation, min and max temperature, distance to city, distance to roads, travel time to city, population count, and population density (Supplementary Table S1). Counterfactuals were assigned where all exact matching was satisfied, and then to the maximum propensity score value from the quantitative covariates. Once a counterfactual pixel was matched to a PA pixel, it was removed from the potential control dataset (without replacement). The propensity scores were based on a logistic regression model, adopted from the methods described in refs. 64–66. We implemented a threshold of a minimum of 5 GEDI shots per 1 km matched pixel to

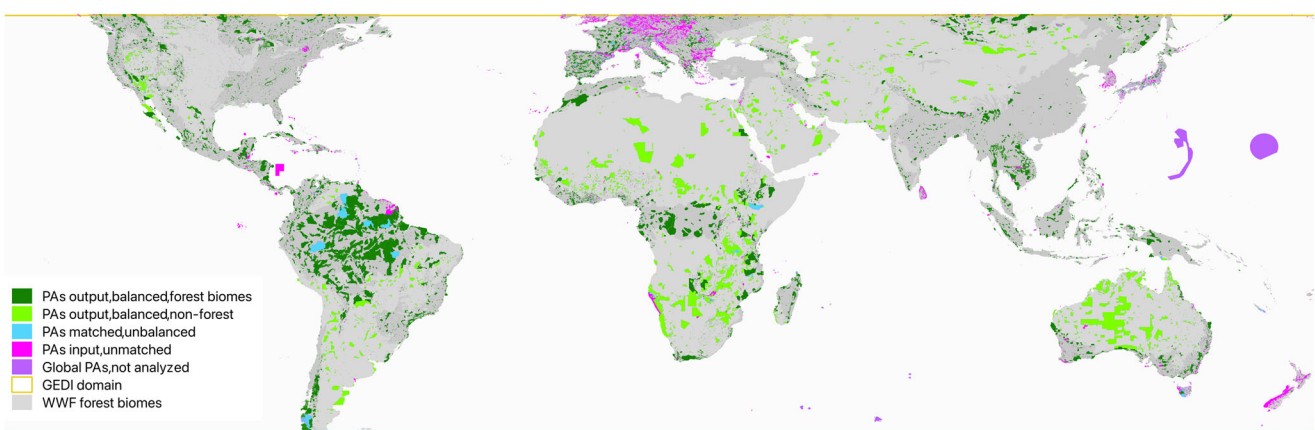

**Fig. 7 | Protected areas analyzed in this study.** Areas with PA-level ecological counterfactuals are shown in green, where PAs without matched counterfactuals are shown in magenta, and PAs with insufficient GEDI data in matched unprotected areas are shown in cyan. For green PAs, the C differences were computed at the individual PA level, for cyan estimates were based on national averages, and for magenta estimates were based on averages from the continental-biome level. World base map made with Natural Earth.

ensure reasonable representation of the forest structure of a given cell. Although the number of GEDI shots in each 1 km pixel varies due to the sampling nature of GEDI (Supplementary Fig. S1), the difference in number of samples per cell does not influence our results as we averaged the structural metrics in all 1 km pixel pairs within each PA and set of PA counterfactual cells for assessing PA effectiveness. The performance of the matching algorithm was evaluated by inspecting the distributions of propensity scores after each step of matching following[54]. To reduce computational time in the propensity score matching, we set a caliper[54,66] to constrain the degree of difference between potential controls and treatment by requiring an overlap in the range of propensity scores for treatment and potential control datasets. This matching algorithm is thoroughly detailed in a complementary study focused on Tanzanian PAs[54].

We assume negligible spatial autocorrelation between PA and counterfactual pixels, as GEDI's AGBD product has been demonstrated to exhibit minimal spatial autocorrelation[67] at its 1-km resolution (each 1-km pixel is comprised of several AGBD samples spaced along track at minimum 60 m). Analyzed PA pixels are also at a minimum 10 km from counterfactuals matches. Other studies have examined the influence of spatial autocorrelation between sets of PA and counterfactual points on the perceived effectiveness of PAs to curb forest loss, and of the approaches explored our matching algorithm most closely resembles the independent matching presented in ref. [68], which was found to be the least affected by spatial autocorrelation. While a thorough analysis of potential spatial autocorrelation between counterfactuals is outside the scope of this study, considering its global scale, further attention to this matter is encouraged for national or subnational studies.

### Global ecosystem dynamics investigation data processing
We used data from GEDI's footprint-level height product (GEDI02_A), cover product (GEDI02_B), and biomass product (GEDI04_A). We applied quality filters to each of these products to only include high-quality forest structure data. We filtered any data where the GEDI02_A quality_flag was equal to 1 and the sensitivity metric was at least 0.95 (i.e., GEDI's lasers were capable of penetrating more than 95% canopy cover for all data used in this study).

We then extracted forest structure metrics from quality-filtered GEDI data from all matched 1 km pixels with GEDI data within PAs and within matched unprotected areas. The total number of filtered GEDI shots used in this analysis was 412,100,767. For each GEDI sample, four structure metrics (RH98, canopy cover, PAI, and AGBD) were aggregated for comparison at a national, continental, and biome level. AGBD

means and totals were converted to AGCD. The difference in mean AGCD between protected areas and matched unprotected counterfactuals was used to indicate the carbon effectiveness of PAs at the given level of aggregation. The distribution of GEDI samples over a single protected area in the Brazilian Amazon, with the associated matched pixels, illustrates the sampling nature of this comparison (Fig. 8).

### Total biomass, uncertainty estimation and carbon expansion
We estimated the mean and total PA biomass (AGBD) and associated uncertainty in each PA, and for the aggregation of all PAs at a country, continent and biome scale using the statistical hybrid inference approach developed for the GEDI mission[24]. This approach accounts for uncertainty resulting from footprint biomass model parameter error and the GEDI sampling design to produce an estimate of the mean biomass and the standard error of the mean at any scale ≥1 km². This algorithm was applied for each PA to estimate a mean PA AGBD, with an associated standard error. The total AGB for the area used the mean multiplied by the non-overlapping PA area, with a confidence interval generated using the standard error of the mean and weighted by the size of each strata. The set of matched 1 km pixels for all PAs and unprotected counterfactuals in each biome and within every country was aggregated. The mean difference and associated standard error of the mean difference was computed at a country-by-biome scale. This mean difference was also multiplied by the PA area to estimate the total additionally preserved AGB attributed to the PA status. Results were aggregated to a country, continent, biome, and global scale by totaling the PA level estimated additionally preserved biomass and associated uncertainties. For unmatched PAs, where there was insufficient GEDI data, we extrapolated from country-by-biome results to estimate the total additionally preserved biomass. If county-by-biome results were unavailable (e.g., in the case of rare entirely protected biomes), extrapolation was from a continent-by-biome level. Conversion of all AGB estimates to carbon used a conversion factor of 0.49, the IPCC global average for conversion factor for woody biomass[69].

### Comparison of year 2000 aboveground carbon density
To test whether our results simply reflect preferential PA establishment in areas of higher AGC, we analyzed a year 2000 AGCD product and assessed whether year 2000 AGCD differed between PAs and matched unprotected areas (insofar as can be gleaned from the only available global year 2000 AGCD product[3]). The results found that the PA age (or time since establishment) was positively correlated with the difference in 2000 AGCD, i.e., that there was a statistically significantly

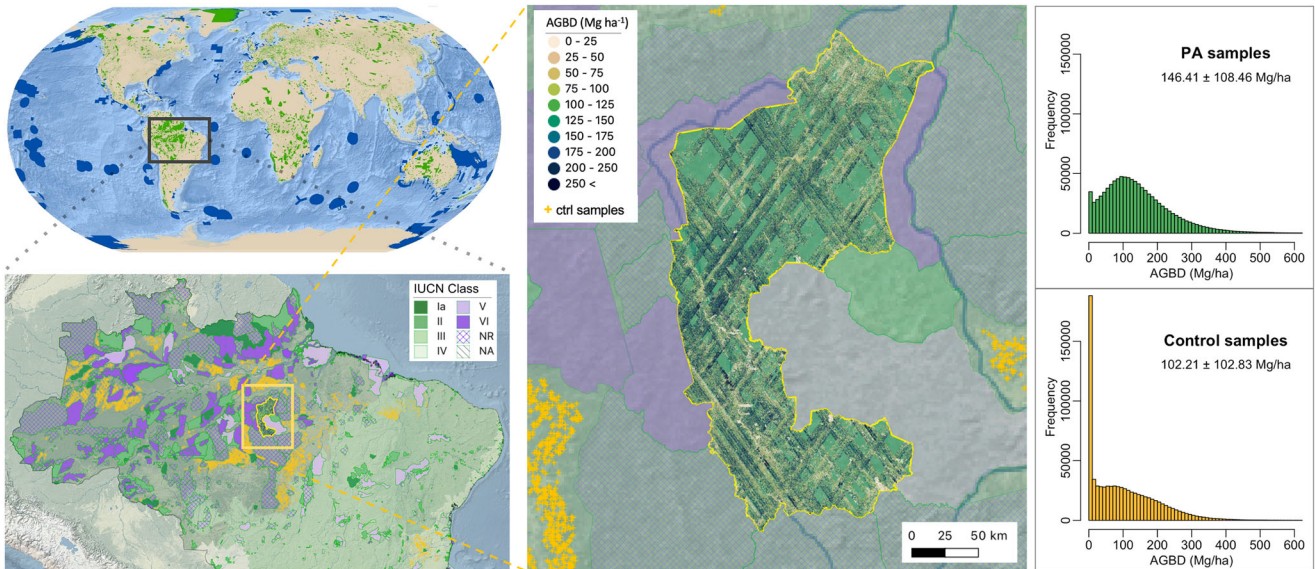

**Fig. 8 | Visualization of matching algorithm.** Each 1 km pixel within protected areas that had GEDI coverage was matched to ecologically similar unprotected 1 km pixels in the same country and biome (shown in yellow for this example from Brazil). GEDI data were then extracted from within each PA and its ecological counterfactuals, and differences in the distributions of height, cover, PAI, and AGB were analyzed (right). World base map made with Natural Earth.

higher ($P < 0.05$) AGCD in PAs than counterfactuals in the year 2000 AGCD product. However, this difference was primarily exhibited in PAs established well before the year 2000 (e.g., >100 years old, Supplementary Fig. S4), where older PAs had larger differences in carbon densities between PAs and counterfactuals in 2000. For PAs designated in and after 2000 (recently established PAs), this difference was significantly smaller (52.7%) than that for PAs designated before the year 2000 (older PAs). By analyzing the relationship between PA age and additionally preserved AGC in the year 2000 (our baseline year), it was inferred that PAs' ability to preserve additional AGC increased over time (Supplementary Fig. S4). These results indicate that at least since the year 2000, PAs have not been preferentially established in higher AGCD sites (i.e., matched unprotected areas are balanced with respect to AGCD). We further divided this analysis into forest and non-forest ecosystems to address whether observed differences were primarily from forest carbon, and found that forested protected areas exhibited higher correlations between PA effectiveness and PA age. Indeed, higher year 2000 AGCD in older forested PAs reinforces our finding that PAs are effective at protecting AGC, and that this effect increases over time as discrepancies between protected and unprotected areas increase.

## Comparison of forest cover loss within and outside of PAs

To test the hypothesis that PAs reduce deforestation or forest loss, we compared the area of forest loss within PAs and their counterfactuals at a country-, biome-, continent- and global level. We first summarized tree cover loss for all forested regions of the globe between years 2000 and 2019 from a Global Forest Change (GFC) data product (v.1.7[25];). The GFC dataset was divided into $10 \times 10°$ tiles with ~30-meter pixel resolution at the equator. We used the GFC "lossyear" data layer to generate a forest cover loss mask layer with values 0 indicating no change in forest cover and values of 1 indicating forest cover loss between 2000 and -2019 at the 30 m pixel resolution. We then calculated the forest cover loss area within every protected area polygon and matched unprotected pixel. We estimated the fraction of forest cover loss (30 m pixels with value 1) falling within each PA and matched unprotected 1 km pixel. We aggregated these values for all the 1 km PA pixels and their corresponding matched unprotected areas to estimate the total forest cover loss differences at the country, biome, continent and global scale.

For the subset of PAs that had matched unprotected areas, we classified the PA into five classes based on differences between PA and unprotected AGB and forest cover loss. Where PAs have higher (>5 Mg/ha) AGBD (positive "carbon effectiveness") than counterfactuals, their effectiveness is attributed either to avoided emissions from forest loss outside of PAs (where there is a higher rate of observed 2000–2000 forest cover loss in counterfactuals than PAs, 30.3% of PAs), or to enhanced stocks (either avoided degradation or enhanced growth, 18.4%) when there is either no difference in forest cover loss between PAs and counterfactuals, or more forest cover loss within PAs. The forest cover loss product is based on Landsat data which saturates with respect to forest cover, height, and biomass, and thus may not be able to detect subtle losses. Where there was little difference in PA and unprotected AGCD (0 +/− 2.5 Mg/ha, 26%), PAs are classified as having no additionality in AGBD, regardless of any perceived difference in forest loss rate. Finally, where PAs had lower AGBD than counterfactuals, this is explained either by encroachment into PAs that is visible as forest cover loss within PAs (8.3%) or degradation in PAs when there is no higher perceived forest cover loss (17%). This analysis was repeated at a continental scale (Supplementary Fig. S5).

## Reporting summary

Further information on research design is available in the Nature Portfolio Reporting Summary linked to this article.

## Data availability

All data used in this study are from publicly available sources. GEDI data are archived on NASA Distributed Active Archive Centers (DAACs). GEDI's footprint-level height data were taken from the GEDI02_A height and elevation product, available at LPDAAC: 10.5067/GEDI/GEDI02_A.002. GEDI's PAI and cover data were taken from the GEDI02_B product also available at LPDAAC: 10.5067/GEDI/GEDI02_B.002. Finally, GEDI's footprint-level biomass (AGBD) data were taken from the GEDI04_A. The WDPA database can be downloaded at www.protectedplanet.net. For the matching variables used in the preprocessing, the 2000 land cover products can be downloaded at http://maps.elie.ucl.ac.be/CCI/viewer/download.php. The WWF ecoregions and biomes can be downloaded at https://www.worldwildlife.org/publications/terrestrial-ecoregions-of-the-world.

The gridded population datasets are retrieved from https://doi.org/10.7927/H4JW8BX5. The annual mean precipitation and temperature datasets are processed from the WorldClim version 1 datasets downloaded from https://developers.google.com/earth-engine/datasets/catalog/WORLDCLIM_V1_MONTHLY#description. Elevation and slope are processed using CGIAR SRTM downloaded from https://developers.google.com/earth-engine/datasets/catalog/CGIAR_SRTM90_V4. Distance to cities dataset is retrieved from https://doi.org/10.3390/land8010014. Travel time to cities dataset can be downloaded from https://forobs.jrc.ec.europa.eu/products/gam/download.php. For additional details related to the matching variables, see supplementary Table 1. Intermediate datasets such as pre-processed matching results are available upon request.

## Code availability

All analysis code is available on GitHub at https://github.com/lauraduncanson/GEDI_PA.git, with the version used for this paper archived at https://zenodo.org/badge/latestdoi/284058541[70].

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

## Acknowledgements

The authors gratefully acknowledge the large number of contributors of field and airborne lidar data that enabled the creation of the empirical biomass models that underpinned the biomass products analyzed in this study. We thank the National Science Foundation for funding through awards 2225078 and 2225076 (B.E., C.M., P.R.R) and OAC1934389 (L.D., and S.M.), and also gratefully acknowledge the continued support of the GEDI mission from the NASA Terrestrial Ecology program, and funding from National Aeronautics and Space Administration (NASA) grants 80NSSC21K0196 (L.D., M.L., and V.L.), 80NSSC19K0186 and 80NSSC21K0189 (S.G.), and NASA contract NNL 15AA03C (R.D. and J.A.). Additionally, S.C. and P.R. acknowledge the funding support received from Betty and Gordon Moore, and John and Jody Arnhold.

## Author contributions

Conceptualization: L.D., L.F., M.L., K.T., and P.R. Methodology: L.D., M.L., V.L., J.A., S.C., M.G., and A.Z. Investigation: L.D., M.L., V.L., and S.M. Visualization: M.L., V.L., and L.D. Funding acquisition: B.E., C.M., and L.D. Writing—original draft: L.D., M.L., V.L., R.D., B.E., L.F., S.G., C.M., P.R., K.T., J.A., S.C., and S.M. Writing—review & editing: L.D., M.L., V.L., R.D., S.M., S.C., B.E., L.F., S.G., C.M., P.R., K.T., and A.Z.

## Competing interests

The authors declare no competing interests.
