## [Peer Review File · Nature Communications]

Reviewer comments

Reviewer #1 (Remarks to the Author):

The authors generate estimates of forest biomass and canopy structure-indices from lidar samples collected by NASA's new GEDI mission and then use them to assess whether protected areas, globally, harbor greater biomass stocks and more complex canopy structure. The study examines a purported phenomenon (additional carbon storage and habitat enhancement within protected areas) that is important to the efficacy of conservation and climate change mitigation (namely, nature based or "natural" climate solutions) but one that, to date, has not been interrogated with an acceptable degree of rigor.

To me, the main advance of this study is its approach to assessing additionality. For the effects of an intervention to be "additional", one needs to somehow show that a given intervention produced a greater outcome than would be expected in its absence. An inherent challenge to this sort of work is establishing a reasonable counterfactual: a defensible prediction of the expected outcome absent the intervention. In the past, this might have been done by simply comparing conditions on either side of a protected areas border or—more advanced—using integrated assessment models that simulate scenarios with and without an intervention to identify where potential land use changes might have been prevented by the intervention. But the former approach doesn't speak to additionality much at all while the latter is imperfect in many other ways: the spatial resolution of land use predictions is often very coarse, carbon stock change estimates are often based on rudimentary assumptions/data, and the underlying economic simulations are based on and thus highly sensitive to myriad assumptions (e.g., <https://doi.org/10.1016/j.jclepro.2022.131477>). To circumvent many of these shortcomings, the authors use a new approach that effectively makes pairwise comparisons between samples within a given protected area and samples taken from outside of the protected area that share similar biophysical and socioeconomic attributes. This approach assumes that forest biomass/structure would be equal among sites sharing similar biophysical and socioeconomic conditions and thus attributes any observed difference to the presence or absence of formal land protection. While still imperfect in the sense that it assumes that all variation can otherwise be explained by a small set of biophysical and socioeconomic covariates, I believe that the approach offers a meaningfully more transparent and intelligible comparison. It also facilitates more detailed estimates of forest biomass/structure than could ever be afforded by the integrated-assessment approach and for follow-on spatial analyses concerning the reasons for observed differences between protected and similar unprotected sites (e.g., are differences explained by observed land use changes, less visible forest degradation, or invisible productivity enhancements?).

I view this as a in important paper that is well suited for publication in Nature Communications. It is the most rigorous analysis of protected area effects on carbon storage that I've seen and is made possible in part by advanced new data from NASA's GEDI mission. The story is simple and straightforward, but no less interesting and compelling. Having said that, I have a few residual concerns with the study that I feel should be addressed before the paper is accepted. These largely pertain to the clarity of the studies methods and results. I outline these in detail below:

Line 27- Please consider clarifying here in the abstract (~one sentence) what you mean by "additional". I first read this as suggesting there was 19.7 Gt biomass in all protected areas and didn't realize that was the additional affect of protected-status until I saw the 125.3 value two sentences later.

Line 27- A general concern I have with the study is exemplified here: You contextualize the study around climate change mitigation and carbon storage/accumulation but the report results in units of 'dry biomass', leaving the reader to make their own unit conversions. Here for example, you report 19.7 Gt of biomass, then refer to it as "these higher *C* stocks..." and then state that their roughly equal to global fossil fuel emissions (which are ~10 GtC)—the risk is that folks might miss the quiet shift in units being discussed and incorrectly assume that global fossil fuel emissions are ~20 GtC. Given the study's context (C storage) I'd personally prefer to see results reported in C

units. But I understand that would be quite a burden. At the very least, please ensure that the units being discussed are clear and note clearly how comparisons are being made.

Line 28- I worry that comparing the 19.7 Gt figure to *annual* fossil fuel emissions could be misinterpreted as saying that C storage/accumulation of protected areas offsets fossil fuel emissions each year. I suggest changing the phrasing here to read something like: "...and are roughly equivalent to just one year's worth of global fossil fuel emissions."

Line 64- To whose "assumption" are you referring here? Given this is essentially the studies motivation, a citation or two seems warranted.

Lines 94-97- Since the methods are (and will be) at the end of the text, I found this passage to be particularly confusing. Based on my understanding of your methods, consider clarifying along the lines of: "We matched PA's to unprotected areas using a suite of ecological, anthropogenic pressure, and climate variables representing conditions in 2000. We therefore assume that forest structure/biomass was equal at that time; By then comparing 2020 GEDI measurements from these protected/unprotected pairs, our results thus represent approximately 20 years of change associated with an areas PA designation."

Line 117- Consider adding a sentence here for less familiar readers describing how protected area status might lead to "enhanced growth".

Line 125- Please cite the 2000 AGB map you used, here.

Table 1- When reporting biomass for biomes like montane, tropical, and/or temperate "grasslands and shrublands", are you just considering the woody biomass (i.e. the shrubs) or is GEDI also picking up herbaceous biomass?

Lines 290-291-This sentence ("Our results highlight synergies in efforts and resources into protected areas..." is unclear.

Line 372- Why 10km? Was a sensitivity test conducted? Or spatial autocorrelation assessed?

Lines 372-374- Do I understand correctly then that the pixels within a given protected area were never compared to the pixels within 10km of *any* protected area? Or were just the pixels within 10km of the given protected area excluded? It seems that if pixels within 10km are being excluded to remove potential spillover-effects, then pixels within 10km of *any* protected area should be excluded. Perhaps this is what you did? Either way, please clarify in the text.

Lines 378-379- You describe both "exact matching" and "propensity score matching" but it's unclear to me which matching method was used to what end in your analyses. Please clarify in the text. This whole section ("Matching Algorithm") feels a bit vague and confusing to me—probably because you're trying to describe a complex analysis in very limited space. Since this particular method is at the core of your study's main contribution (assessing additionality) I'd like to see a more detailed description. If space is the limitation, consider including part of that description in the supplement or moving bulk of the less important methods (i.e., the meat of the GEDI and IUCN dataset descriptions to the supplement).

Line 380- The meaning of the parenthetical "(one to one matching)" is unclear to me.

Lines 381-384- Please reference Table S1 at the end of this sentence. You reference it in the topic sentence of this paragraph, but I personally feel it would be better situated here where you're actually discussing the specific data used.

Lines 387-389- So, what was the range (the "caliper"?) of the propensity scores considered acceptably equivalent? Why?

Lines 629-644- This information may be worth including in the main text since it's central to your assumption of uniformity in 2000 and speaks further to the efficacy of protected areas.

Line 634- Please be more specific than "heavily influenced". In what direction? To what magnitude?

Line 635- Again, please be more specific. "Statistically significant difference" doesn't tell me which land designation harbored greater C density. Consider adding greater detail along the lines of: "PA's had significantly larger [or smaller?] C stocks than unprotected areas in 2000..."

Line 639: "it was inferred"—how? Please elaborate. E.g., "we observed that older PA's harbored greater biomass densities than younger PA's (Figure S3)."

Figure S3: What is the significance of the partition between non-/forest-dominated biomes? I do not recall this being discussed in the context of protected area age.

Figures S4 & S5: Nice figures! How was the cross-hatched region ("No additionality in biomass...") determined? Is it +/- 5Mg/ha? Please make this clearer in the caption.

Reviewer #2 (Remarks to the Author):

Major general comments

I thoroughly enjoyed reading this paper, thank you. The manuscript assesses the extent to which protected areas have effectively avoided forest carbon loss (aboveground biomass). The question has important ramifications for nature-based solutions to global climate change and will be of interest to many readers. Overall, this is an interesting subject, however the paper could be improved with some minor tightening of the storyline, defining and explaining some generalised claims, and considering the regional results in light of the deforestation pressures they face, where room permits.

Specific comments

Review caps versus small case. For some terms, they change throughout. Story could be strengthened throughout by restructuring paragraphs into key themes. Avoid jargon where possible. Or define when necessary.

Introduction:

Line 49: 'Forest conservation is a crucial mechanism for forest management, climate change mitigation, and curbing biodiversity loss' Briefly clarify why, particularly for forest management it is not obvious. Previous paragraph explains role in carbon cycle but other reasons are new to the story.

Line 51: Without room to explain detail on how PAs are key indicators for achieving conservation goals, perhaps say they are important for 'reporting' on biodiversity conservation goals.

Line 51-52: 'While most efforts to establish protected areas have been focused on biodiversity protection, there are clear co-benefits of biodiversity and carbon conservation efforts' consider rewording to 'there are clear carbon co-benefits'.

Line 53. Protected areas effectively avoid forest cover loss.' is a big generalisation. It needs to be qualified. E.g. Relative to... Depends on area... Otherwise consider rewording to something like 'When protected areas effectively avoiding forest loss, they also provide carbon sequestration and climate mitigation benefits.'

Line 79: Counterfactuals is more easily understood than counterparts. A suggestion only.

Results

Line 118-119. 'For these 18% of PAs without apparent forest loss in matches...' consider rewording to help the reader understand

Discussion

Line 247-248: 'climate forcing' 'forest albedo' reword or define

Lines 259-259: This is interesting to explore further. This is a global study and therefore will not be able to dive into regional or local trends, but this is ultimately the crux of the rationale and it would be interesting to read more about the authors interpretation of this result. Please see a paper focused on Southeast Asia, which may provide some background:

Graham, V. et al. (2021) Southeast Asian protected areas are effective in conserving forest cover and forest carbon stocks compared to unprotected areas. *Scientific Reports* 11, 23760, <https://www.nature.com/articles/s41598-021-03188-w#Sec5>

Line 275-276: This is a potentially contentious claim. Suggest either explain and justify or remove.

Methods:

Did you explore correlation between covariates used in the matching? If so, explain.

Did you consider the effect of spatial autocorrelation on the results? If so, explain.

See

Negret, P. J. et al. Effects of spatial autocorrelation and sampling design on estimates of protected area effectiveness. *Conserv. Biol.* 34, 1452–1462. <https://doi.org/10.1111/cobi.13522> (2020).

And Schleicher, J. et al. Statistical matching for conservation science. *Conserv. Biol.* 34, 538–573 549 (2020).

Reviewer #3 (Remarks to the Author):

This is a significant and important paper. I believe it is the first statistically rigorous paper to assess the global impact of the world's protected area dataset on their carbon stocks, and is an innovative and important use of the GEDI spaceborne LiDAR dataset. The implications go far beyond the field of remote sensing and conservation, giving good reason for it to be published in a non-specialist journal. The figures will be important for NGOs and policy makers, and will be widely circulated.

The new evidence presented for reduced DEGRADATION rates within PAs compared to matched similar pixels in the year 2000 is important, and has not been quantitatively shown on such scale before.

Overall I find this to be a sound paper, with evidence that supports the conclusions, and no major and obvious flaws. I do have some concerns though, which I think should be fixed or clarified prior to publication.

GENERAL POINTS

GEDI does not cover the whole world, only going up to 51 degrees north/south (about the latitude of London). I do not think this is a big problem, and I do not suggest the analysis is changed. But the 'global' in the title and implied in the abstract is incorrect, as the numbers miss a number of protected areas in Canada, Russia, northern Europe (e.g. most of the UK, Scandinavia). This needs to be made clear. This could lead to the figures from the paper being used in an inaccurate way.

Similarly Figure 2 is misleading. By showing land in northern Europe (say) along with a graph saying 'Europe', but based on data that does not extend to the northern half of Europe, the reader could be misled. I suggest cropping the world map in this figure and making it much clearer to the reader where the data are incomplete.

It would be useful I think for a general reader to have more understanding given in the Introduction and Results about how GEDI functions. For example pointing out its sampling nature, and saying what proportion of each PA is actually sampled by GEDI shots, and how it varies by cloud cover and orbit patterns. A figure showing the % of the average PA in a country that is

covered by GEDI shots might be useful for example? This might show more missing data in (say) Indonesia compared to less cloudy places.

Similarly, a little bit of an introduction to matching, and brief table or similar saying what datasets you used for matching, I think belongs in the main text of the paper (Introduction maybe?) not in the Methods only. Fig 6 is good, but some info on datasets could be included there, there is space.

I am not an expert in matching algorithms - from what I can see the methods seem sensible. I have two concerns though:

Was any effort made to ensure a similar number of GEDI shots per matching 1 km pixels in/out PAs?

1km is a coarse resolution to deal with deforestation and degradation, as things in/out the PA could get confused. Were efforts made to ensure that every PA 1km pixel was 100% within a PA? If not, I do have worries about the whole analysis - but from my reading of the methods there is no concern here as they state that efforts were made to avoid PA boundaries so I think this was considered. And similar, I assume PAs smaller than a few 1 km pixels were excluded?

MINOR POINTS

Abstract talks about 'C stocks' but then gives numbers in terms of AGB. Maybe simpler for the reader to convert numbers to C in abstract?

First sentence in text switches between AGB and Gt CO₂. Again, I would prefer a single unit throughout to avoid confusion (non-forest biomass scientists do not have the conversions between biomass, C, and CO₂ ready to use in their heads as we do!) Maybe just doing everything in Mg C throughout would help?

I dislike the comparison in the abstract to "are roughly equivalent to annual global fossil fuel emissions". This is conflating a total stock with an annual flux. Would be better phrased as "this extra stock of carbon is roughly equivalent to a single year of global fossil fuel emissions".

Figure of 26% of all AGB being in protected areas is higher than 14.5% reported for the tropics in Collins & Mitchard 2017 <https://www.nature.com/articles/srep41902> Could mention this discrepancy in the paper, and explain whether due to new protected areas since then, tropics having a lower proportion protected than non-tropics, or different AGB dataset? Note there is vague mention of previous studies line 66-67, which says they fail as previous AGB datasets saturate in high biomass forest - but this is not always true, the Collins & Mitchard paper used the Avitabile et al. 2016 carbon map which does not saturate.

Fig 1 B-E is incorrectly cropped, with an extra strip with no date covering e.g. the UK included when it is not in 1A. I think these should be cropped to match 1A

Not sure Fig 4 parcels C/D are that useful - largely a function of the area of the countries. Same things done per unit area might be useful?

Line 298 - REDD+ acronym wrongly spelled out: REDD+ is "Reducing emissions from deforestation and forest degradation, conservation of existing forest carbon stocks, sustainable forest management and enhancement of forest carbon stocks" according to relevant UNFCCC texts - not sure REF 55 relevant here. I don't think this dataset necessarily helps with including protected areas under voluntary sector REDD+ projects - the fact that they're doing better does not prove additionality on its own. But also under UNFCCC REDD+ there's no reason they can't be in anyway ("conservation of existing carbon stocks"). Suggest rewrite these couple of sentences.

Reviewer #1 (Remarks to the Author):

The authors generate estimates of forest biomass and canopy structure-indices from lidar samples collected by NASA's new GEDI mission and then use them to assess whether protected areas, globally, harbor greater biomass stocks and more complex canopy structure. The study examines a purported phenomenon (additional carbon storage and habitat enhancement within protected areas) that is important to the efficacy of conservation and climate change mitigation (namely, nature based or "natural" climate solutions) but one that, to date, has not been interrogated with an acceptable degree of rigor.

To me, the main advance of this study is its approach to assessing additionality. For the effects of an intervention to be "additional", one needs to somehow show that a given intervention produced a greater outcome than would be expected in its absence. An inherent challenge to this sort of work is establishing a reasonable counterfactual: a defensible prediction of the expected outcome absent the intervention. In the past, this might have been done by simply comparing conditions on either side of a protected areas border or—more advanced—using integrated assessment models that simulate scenarios with and without an intervention to identify where potential land use changes might have been prevented by the intervention. But the former approach doesn't speak to additionality much at all while the latter is imperfect in many other ways: the spatial resolution of land use predictions is often very coarse, carbon stock change estimates are often based on rudimentary assumptions/data, and the underlying economic simulations are based on and thus highly sensitive to myriad assumptions (e.g., <https://doi.org/10.1016/j.jclepro.2022.131477>). To circumvent many of these shortcomings, the authors use a new approach that effectively makes pairwise comparisons between samples within a given protected area and samples taken from outside of the protected area that share similar biophysical and socioeconomic attributes. This approach assumes that forest biomass/structure would be equal among sites sharing similar biophysical and socioeconomic conditions and thus attributes any observed difference to the presence or absence of formal land protection. While still imperfect in the sense that it assumes that all variation can otherwise be explained by a small set of biophysical and socioeconomic covariates, I believe that the approach offers a meaningfully more transparent and intelligible comparison. It also facilitates more detailed estimates of forest biomass/structure than could ever be afforded by the integrated-assessment approach and for follow-on spatial analyses concerning the reasons for observed differences between protected and similar unprotected sites (e.g., are differences explained by observed land use changes, less visible forest degradation, or invisible productivity enhancements?).

I view this as an important paper that is well suited for publication in Nature Communications. It is the most rigorous analysis of protected area effects on carbon storage that I've seen and is

made possible in part by advanced new data from NASA's GEDI mission. The story is simple and straightforward, but no less interesting and compelling. Having said that, I have a few residual concerns with the study that I feel should be addressed before the paper is accepted. These largely pertain to the clarity of the studies methods and results. I outline these in detail below:

Thank you very much for the positive feedback and helpful synthesis of our algorithmic approach for creating counterfactuals. We have endeavored to address all of your (very helpful) comments below. Responses in blue, edits to the paper in red.

Line 27- Please consider clarifying here in the abstract (~one sentence) what you mean by "additional". I first read this as suggesting there was 19.7 Gt biomass in all protected areas and didn't realize that was the additional affect of protected-status until I saw the 125.3 value two sentences later.

Thank you for this - we have restructured the abstract to first present the total AGB (now C) numbers in PAs, followed by the specific attribution to PA status: 'Here we used ~412 million lidar samples from NASA's GEDI mission to estimate a total of The total measured PA AGB of 61.4.3 GtC (+/- 0.31), 26% of all mapped terrestrial woody carbon. Of this total, 9.6519.7 +/- 0.881.8 Gt of additional carbonAboveground Biomass (AGB) was attributed toassociated with PA status. These higher (additional) C stocks are primarily fromattributed to avoided emissions from deforestation and degradation in PAs compared to unprotected forests This total isand are roughly equivalent to one year of annual global fossil fuel emissions. The total measured PA AGB was 125.3 Gt (+/- 0.63), 26% of all mapped terrestrial woody AGB. These results underscore the importance of conservation of high biomass forests for avoiding carbon emissions and preserving future sequestration.'

Line 27- A general concern I have with the study is exemplified here: You contextualize the study around climate change mitigation and carbon storage/accumulation but the report results in units of 'dry biomass', leaving the reader to make their own unit conversions. Here for example, you report 19.7 Gt of biomass, then refer to it as "these higher *C* stocks..." and then state that their roughly equal to global fossil fuel emissions (which are ~10 GtC)—the risk is that folks might miss the quiet shift in units being discussed and incorrectly assume that global fossil fuel emissions are ~20 GtC. Given the study's context (C storage) I'd personally prefer to see results reported in C units. But I understand that would be quite a burden. At the very least, please ensure that the units being discussed are clear and note clearly how comparisons are being made.

Thank you for this feedback - another reviewer shared this same concern. For clarity of communication and consistency in the writing we have changed all of the biomass numbers to carbon numbers throughout the text, tables and figures. This was relatively simple as we simply

used the IPCC mean conversion factor of 0.49. All means are now in C Mg/ha and totals are in GtC.

Line 28- I worry that comparing the 19.7 Gt figure to *annual* fossil fuel emissions could be misinterpreted as saying that C storage/accumulation of protected areas offsets fossil fuel emissions each year. I suggest changing the phrasing here to read something like: "...and are roughly equivalent to just one year's worth of global fossil fuel emissions."

Thank you and we completely agree, and have rephrased accordingly: 'This total is roughly equivalent to one year of annual global fossil fuel emissions.'

Line 64- To whose "assumption" are you referring here? Given this is essentially the studies motivation, a citation or two seems warranted.

Thanks and fair enough! We have added what we feel is the most relevant citation here from Melilo et al., 2016.

Lines 94-97- Since the methods are (and will be) at the end of the text, I found this passage to be particularly confusing. Based on my understanding of your methods, consider clarifying along the lines of: "We matched PA's to unprotected areas using a suite of ecological, anthropogenic pressure, and climate variables representing conditions in 2000. We therefore assume that forest structure/biomass was equal at that time; By then comparing 2020 GEDI measurements from these protected/unprotected pairs, our results thus represent approximately 20 years of change associated with an areas PA designation."

Thank you for this point, and especially for providing such a succinct suggestion for rephrasing. We have followed your advice and rephrased as: 'PAs were matched to unprotected areas using a suite of ecological, anthropogenic pressure, and climate variables representing conditions in 2000. We therefore assume biomass densities in our samples within PAs and counterfactuals were equal at that time. By then comparing 2020 GEDI measurements between these protected/unprotected pairs, differences in 2020 represent approximately 20 years of change associated with PA status.'

Line 117- Consider adding a sentence here for less familiar readers describing how protected area status might lead to "enhanced growth".

Thanks for your comment - we have slightly rephrased as enhanced regrowth rather than enhanced growth - the citations in the following sentence both provide evidence of faster regrowth in PAs compared to unprotected forests, while the Mills et al. citation shows that degraded forest regrowth is slower than previous studies have found. 'Although we attribute PAs

where we see higher AGB without reduced forest cover losses as avoided degradation, PA vegetation in these cases may also be exhibiting enhanced growth regrowth compared to unprotected forests (ii) (Melillo et al. 2016; Mills et al. 2023). Our assertion of enhanced regrowth is supported by local and regional studies assessing PA forest growth^{28,29}.’

Line 125- Please cite the 2000 AGB map you used, here.

Response: thank you for pointing this out, the product citation is added both in text and in the reference list.

Citation: Harris, N.L., D.A. Gibbs, A. Baccini, R.A. Birdsey, S. de Bruin, M. Farina, L. Fatoyinbo, M.C. Hansen, M. Herold, R.A. Houghton, P.V. Potapov, D. Requena Suarez, R.M. Roman-Cuesta, S.S. Saatchi, C.M. Slay, S.A. Turubanova, A. Tyukavina. 2021. Global maps of twenty-first century forest carbon fluxes. *Nature Climate Change*. <https://doi.org/10.1038/s41558-020-00976-6>

Table 1- When reporting biomass for biomes like montane, tropical, and/or temperate “grasslands and shrublands”, are you just considering the woody biomass (i.e. the shrubs) or is GEDI also picking up herbaceous biomass?

GEDI’s biomass products are only representative of aboveground woody biomass, not herbaceous biomass. Therefore in these non-forested ecosystems we are only accounting for trees and shrubs. We have added the following note to the Table 1 caption: ‘Note GEDI’s biomass products only account for aboveground woody C, even in non-forest ecosystems (i.e. trees and shrubs, not herbaceous C or soil C stocks).’

Lines 290-291-This sentence (“Our results highlight synergies in efforts and resources into protected areas...” is unclear.

Agreed. Too many co-authors in the kitchen, this seems to have slipped through the last editorial rounds. This sentence has been removed, as the following sentence is clearer and captures the point.

Line 372- Why 10km? Was a sensitivity test conducted? Or spatial autocorrelation assessed?

Response: thank you for this question. This 10 km buffer was determined by surveying the literature, and is used to account for both positive and negative spillover effects as well as encroachment events at the PA boundaries (Liang et al., 2023; Joppa and Pfaff, 2011). 10km is often used in PA impact assessment to be conservative in impact assessment, as including buffers decreased apparent PA effectiveness that may have been misattributed to spillover effects (Joba & Pfaff, 2010, Alexandre et al., 2010). This would also account for potential leakage into

the areas directly surrounding PAs. Knorn et al., 2012, conducted an analysis of different buffer zones, and concluded at least 5 km should be used.

Regarding autocorrelation, we think that's a separate issue - the 1 km resolution of GEDI aggregated samples is outside the range of spatial autocorrelation of GEDI shots (i.e. there should be negligible spatial autocorrelation even when 1 km cells directly neighbor each other). Therefore this should not impact the buffer selected. We did not conduct a sensitivity analysis as 10 km is so frequently used in the literature and we felt this was ample and would produce conservative results for this study.

We have updated the text to justify the 10 km buffer more:

A 10 km buffer around any PA borders was also implemented to remove potential spillover effects e.g. encroachment at the PA boundaries or potential leakage into neighboring forests (Joppa & Pfaff, 2010, Alexandre et al., 2010, Knorn et al., 2012).

Additionally for full transparency we have added a paragraph regarding spatial autocorrelation:

'We assume negligible spatial autocorrelation between PA and counterfactual pixels, as GEDI's AGBD product has been demonstrated to exhibit minimal spatial autocorrelation⁶⁶ at its 1 km resolution (each 1 km pixel is comprised of several AGBD samples spaced along track at minimum 60 m). Analyzed PA pixels are also at minimum 10 km from counterfactuals matches. Other studies have examined the influence of spatial autocorrelation between sets of PA and counterfactual points on the perceived effectiveness of PAs to curb forest loss, and of the approaches explored our matching algorithm most closely resembles the independent matching presented by⁶⁷, which was found to be the least affected by spatial autocorrelation. While a thorough analysis of potential spatial autocorrelation between counterfactuals is outside the scope of this study, considering its global scale, further attention to this matter is encouraged for national or subnational studies.'

Lines 372-374- Do I understand correctly then that the pixels within a given protected area were never compared to the pixels within 10km of *any* protected area? Or were just the pixels within 10km of the given protected area excluded? It seems that if pixels within 10km are being excluded to remove potential spillover-effects, then pixels within 10km of *any* protected area should be excluded. Perhaps this is what you did? Either way, please clarify in the text.

Response: your understanding is correct that pixels within 10km buffering distance from any PAs were excluded in the comparison/matching/analysis. The step is indeed done to avoid including any spillover-effects and is included in the preprocessing procedure. We have added clarification to the text: 'Therefore no locations within the 10 km buffer of any protected area were considered for matching.'

Lines 378-379- You describe both “exact matching” and “propensity score matching” but it’s unclear to me which matching method was used to what end in your analyses. Please clarify in the text. This whole section (“Matching Algorithm”) feels a bit vague and confusing to me—probably because you’re trying to describe a complex analysis in very limited space. Since this particular method is at the core of your study’s main contribution (assessing additionality) I’d like to see a more detailed description. If space is the limitation, consider including part of that description in the supplement or moving bulk of the less important methods (i.e., the meat of the GEDI and IUCN dataset descriptions to the supplement).

Response: thank you for pointing this out. We apologize for the description on the matching method being unclear, we agree that the writing could be clarified. Additionally, during this review period, a complementary paper has been published which details the matching algorithm and its proof of concept in Tanzania. This can now be referenced here: (<https://doi.org/10.1016/j.gloenvcha.2022.102621>). To provide a quick summary, the overall matching mechanism we used in terms “one-to-one matching” where each 1km cell from the protected/treatment group is matched to only one 1 km cell from the unprotected area (control) as compared to multiple in non one-to-one matching. The specific matching method to achieve one-to-one matching includes propensity score matching and exact matching. Propensity score matching selects the best matched control with the highest propensity score calculated using logistic regression models built with geophysical, climatic, and social variables that are continuous. Exact matching requires potential matched control cells to be in the same categorical variables, and is applied to filter out candidates prior to applying propensity score matching.

We have restructured this section and added the citation to the more detailed paper outlining the methods:

‘By comparing the vegetation structure in PAs and their matched unprotected areas, we addressed the fact that PAs were usually non-randomly distributed and assessed the conservation strategy’s efficacy in storing biomass and carbon. To assign counterfactual 1 km pixels to PA 1 km pixels, we required exact matching of several covariates: land cover category, country, ecoregion and biome. In addition, we used propensity score matching for quantitative covariates: geophysical, climatic and social variables, in elevation, slope, mean precipitation, min and max temperature, distance to city, distance to roads, travel time to city, population count, and population density (Table S1). Counterfactuals were assigned where all exact matching was satisfied, and then to the maximum propensity score value from the quantitative covariates. Once a counterfactual pixel was matched to a PA pixel, it was removed from the potential control dataset (without replacement). The propensity scores were based on a logistic regression model, adopted from the methods described in ^{59–61}. We evaluated our matching algorithm performance by inspecting the distributions of propensity scores after each step of matching following ⁵¹. To reduce computational time in the propensity score matching, we set a caliper to constrain the degree of difference between potential controls and treatment by requiring an overlap in the range of propensity scores for treatment and potential control datasets. This matching algorithm is thoroughly detailed in a complementary study focused on Tanzanian PAs (Liang et al. 2023).’

Line 380- The meaning of the parenthetical “(one to one matching)” is unclear to me.

Response: this is hopefully addressed in the response to the comment above, but has been removed from the restructured paragraph.

Lines 381-384- Please reference Table S1 at the end of this sentence. You reference it in the topic sentence of this paragraph, but I personally feel it would be better situated here where you’re actually discussing the specific data used.

Response: we agree with the reviewer on the placement of reference to Table S1 and have changed the location to the end of sentence on L384.

Lines 387-389- So, what was the range (the “caliper”?) of the propensity scores considered acceptably equivalent? Why?

Response: thanks for pointing this out. By setting a range on the propensity score, which is the same concept as setting a caliper (Schleicher et al., 2020), we were essentially setting a bound on the potential controls to be matched to our treatment cells based on the controls’ propensity score values. This step was implemented to filter out controls that have large differences in propensity score values from our treatment dataset, and to save computational resources. The figure below is from Fig 7 of Liang et al. (2023), B and C demonstrated the importance of setting a caliper for propensity score filtering.

We have added the citations to both the Liang et al., and Schleicher et al. 2020 papers, and hope that in addition to the restructured text above this will sufficiently clarify the methods for readers.

Lines 629-644- This information may be worth including in the main text since it's central to your assumption of uniformity in 2000 and speaks further to the efficacy of protected areas.

Thanks for the comment, and upon review we agree, and **have moved this section into the main methods text.**

Line 634- Please be more specific than “heavily influenced”. In what direction? To what magnitude?

Response: We agree this wording is not statistically useful to readers, and have replaced the term ‘heavily influenced’ with ‘positively correlated’. Essentially this statement (and associated analysis) explains that where there are different values of 2000 AGB between PAs and counterfactuals, these are in PAs that were established well before 2000 and thus already had the opportunity for PA status to influence AGB. We have edited the entire paragraph for clarity and hope that the revised version is clearer - we feel it is:

‘Comparison of year 2000 AGBD

To test whether our results simply reflect preferential PA establishment in areas of higher AGB, we analyzed a year 2000 AGBD product and assessed whether year 2000 AGBD differed between PAs and matched unprotected areas (insofar as can be gleaned from the only available global year 2000 AGBD product ⁴). The results found that the PA age (or time since establishment) was **positively correlated with the difference in 2000 AGBD, i.e. that there was a statistically significantly higher ($p < 0.05$) AGBD in PAs than counterfactuals in the year 2000 AGBD product. However this difference was primarily exhibited in PAs established well before the year 2000 (e.g. >100 years old, Fig S3), where older PAs had larger differences in carbon density between PAs and counterfactuals in 2000.** For PAs designated in and after 2000 (recently established PAs), this difference was significantly smaller (52.7%) than that for PAs designated before the year 2000 (older PAs). By analyzing the relationship between PA age and additionally preserved AGB in the year 2000 (our baseline year), it was inferred that PAs’ ability to preserve additional AGB increased over time (Fig S3). These results indicate that at least since the year 2000, PAs have not been preferentially established in higher AGBD sites (i.e. matched unprotected areas are balanced with respect to AGBD). Indeed, higher year 2000 AGBD in older PAs reinforces our finding that PAs are effective at protecting AGB, and that this effect increases over time as discrepancies between protected and unprotected areas increase.

Line 635- Again, please be more specific. "Statistically significant difference" doesn't tell me which land designation harbored greater C density. Consider adding greater detail along the lines of: "PA's had significantly larger [or smaller?] C stocks than unprotected areas in 2000..."

Please see edits above that address this useful comment - thank you!

Line 639: “it was inferred”—how? Please elaborate. E.g., “we observed that older PA’s harbored greater biomass densities than younger PA’s (Figure S3).”

Please see edits above that address this as well.

Figure S3: What is the significance of the partition between non-/forest-dominated biomes? I do not recall this being discussed in the context of protected area age.

Good question - we wanted to see whether the observed differences between PA age and year 2000 AGBD was driven by forested or non-forested ecosystems. We found that, as with most of the paper’s results, forested PAs were driving this relationship, which was unsurprising (max carbon densities outside forests will remain low regardless of the age of PA status).

We have added the following: **‘We further divided this analysis into forest and non-forest ecosystems to address whether observed differences were primarily from forest carbon, and found that forested protected areas exhibited higher correlations between PA effectiveness and PA age. Indeed, higher year 2000 AGBD in older forested PAs reinforces our finding that PAs are effective at protecting AGB, and that this effect increases over time as discrepancies between protected and unprotected areas increase.’**

Figures S4 & S5: Nice figures! How was the cross-hatched region (“No additionality in biomass...”) determined? Is it +/- 5Mg/ha? Please make this clearer in the caption.

Thanks for your comment - yes we simply used 5 Mg/ha, this has been clarified in the caption, although it is now 2.5 Mg/ha C after the paper-wide conversion of AGBD -> AGCD.

Reviewer #2 (Remarks to the Author):

Major general comments

I thoroughly enjoyed reading this paper, thank you. The manuscript assesses the extent to which protected areas have effectively avoided forest carbon loss (aboveground biomass). The question has important ramifications for nature-based solutions to global climate change and will be of interest to many readers. Overall, this is an interesting subject, however the paper could be improved with some minor tightening of the storyline, defining and explaining some generalised claims, and considering the regional results in light of the deforestation pressures they face, where room permits.

Thank you very much for your positive feedback. We have addressed each of your comments individually and feel the modified manuscript is clearer, and hopefully addresses all of your concerns.

Specific comments

Review caps versus small case. For some terms, they change throughout.

We have read through the paper to resolve discrepancies between lower and upper case text, particularly with respect to carbon vs. C, AGBD and AGCD, protected areas and PAs.

Story could be strengthened throughout by restructuring paragraphs into key themes. Avoid jargon where possible. Or define when necessary.

We hope that the revised paper reads more clearly - we feel that paragraphs were already separated through the results section into key themes, and the discussion is short enough that themes seem unnecessary, but we have edited throughout so the storyline will hopefully now be clearer. Regarding jargon, we have also attempted to remove or define any terms that may be considered jargon for the wide readership of Nature Communications.

Introduction:

Line 49: 'Forest conservation is a crucial mechanism for forest management, climate change mitigation,⁴⁹ and curbing biodiversity loss ^{7,8.}' Briefly clarify why, particularly for forest management it is not obvious. Previous paragraph explains role in carbon cycle but other reasons are new to the story.

Thank you for this comment - this was meant to read forest management TOWARD CC mitigation & curbing biodiversity loss, we appreciate you pointing this out. This has been replaced to read: 'Forest conservation is a crucial mechanism for forest management toward climate change mitigation, and for curbing biodiversity loss'

Line 51: Without room to explain detail on how PAs are key indicators for achieving conservation goals, perhaps say they are important for 'reporting' on biodiversity conservation goals.

Thanks, agreed, rephrased as: 'Protected areas are a foundation for global forest conservation efforts and monitoring PA effectiveness is key for determining progress in achieving the UN SDGs'

Line 51-52: 'While most efforts to establish protected⁵¹ areas have been focused on biodiversity protection ¹⁰, there are clear co-benefits of biodiversity⁵² and carbon conservation efforts' consider rewording to 'there are clear carbon co-benefits'.

While we would like to keep the term co-benefits, for clarity we have added the following:

‘While most efforts to establish protected areas have been focused on biodiversity protection ¹⁰, there are clear co-benefits of biodiversity and carbon conservation efforts, **as older, biodiverse forests also typically store more carbon**’

Line 53. Protected areas effectively avoid forest cover loss..’ is a big generalisation. It needs to be qualified. E.g. Relative to... Depends on area... Otherwise consider rewording to something like ‘When protected areas effectively avoiding forest loss, they also provide carbon sequestration and climate mitigation benefits.’

Thank you, agreed. We have rephrased to make the statement more general and matched to the references as ‘Protected areas have been demonstrated to effectively avoid forest cover loss in many regions ^{12,13,14}, as well as regulate temperature and local climate ¹⁵, and potentially boosting carbon sequestration capacity ^{16,17}. Therefore, PA expansion may be a pathway to bolster climate change mitigation¹⁸’

Line 79: Counterfactuals is more easily understood than counterparts. A suggestion only.

Agreed, we have replaced throughout. We have also replaced ‘matches’ with ‘counterfactuals’ for consistency.

Results

Line 118-119. ‘For these 18% of PAs without apparent forest loss in matches...’ consider rewording to help the reader understand

Thank you - essentially we’re saying we can’t definitively attribute the signal in those PAs to avoided degradation or regrowth. For clarity we have stressed this (and the desire for attribution if field work were possible): ‘Based on these results, we cannot definitively attribute the signal in this 18% of PAs to avoided degradation, enhanced growth, or a combination of both. Regardless, there is a clear signal of higher C densities here, and attribution of higher C in these PAs would benefit from further investigation with conservation practitioners, including fieldwork.’

Discussion

Line 247-248: ‘climate forcing’ ‘forest albedo’ reword or define

Rephrased as: Our results highlight the critical importance of Protected Areas to help mitigate climate change. Aboveground carbon stock flux is only one way forests influence climate change, while forest loss also influences albedo, evaporative flux, belowground biomass etc.

Lines 259-259: This is interesting to explore further. This is a global study and therefore will not be able to dive into regional or local trends, but this is ultimately the crux of the rationale and it

would be interesting to read more about the authors interpretation of this result. Please see a paper focused on Southeast Asia, which may provide some background:
Graham, V. et al. (2021) Southeast Asian protected areas are effective in conserving forest cover and forest carbon stocks compared to unprotected areas. Scientific Reports 11, 23760,
<https://www.nature.com/articles/s41598-021-03188-w#Sec5>

Thank you - we totally agree that this is one of the most interesting aspects of the paper - a full ~25% of PAs exhibited no difference in C stocks, but we do not have the data to further understand this. The paper you cited gives further strength to the argument that where background forest loss is highest, we see larger relatively carbon stock in PAs. We have added that citation to the reference to Brazil, accordingly: ‘The correspondence between high effectiveness of PAs for AGB preservation in countries with high forest loss rates (e.g. Brazil⁴⁷, Southeast Asia(Graham et al. 2021))’. That said, we do not want to add further thoughts regarding the ‘ineffective’ PAs as we feel they would be speculative without more data (and as you point out, regionally or nationally specific data) to further explain the results. Please note we do have another paper that has been published recently (Liang et al., 2022) that uses data from Tanzania to understand which PA designations are most effective, and we hope to see more such research over the next few years. We hope these global results will help incentivize these more intensive regional studies.

Line 275-276: This is a potentially contentious claim. Suggest either explain and justify or remove.

Agreed, thank you. We have rephrased as: ‘Aboveground carbon stock flux is only one way forests influence climate change, while forest loss also influences albedo, evaporative flux, belowground biomass etc. which are also likely impacted by protected status.’

Methods:

Did you explore correlation between covariates used in the matching? If so, explain.

Response: this is a very good question, thank you. In brief, based on the statics literature, the correlation between the covariates are not influential to the accuracy of our treatment group assignment or the matching results.

The 12 covariates are used to construct the logistic model for estimating propensity scores, which is defined as the conditional probability of assigning a unit to a particular treatment condition (i.e., likelihood of receiving treatment), given a set of observed covariates (Holmes, 2014; Rubin, 1979). Therefore, the propensity score model or the selection model is different from the outcome model which estimates the effects of the treatment on the outcome variable (in our case AGBD). This difference therefore indicates with the propensity score estimation, our main concern is not with the parameter estimates of the model, but rather with the resulting balance of the covariates (Augurzky and Schmidt, 2001), therefore standard concerns

about collinearity do not apply (Stuart, 2010). Secondly, to achieve balance of the covariates or to equate the distribution of propensity scores in the treated and control groups, literature has indicated that the propensity score model is robust against including more variables that may be associated with the treatment assignment, while accidentally excluding potentially important confounder can be very costly in terms of increased bias (Stuart, 2010; Rosenbaum & Rubin, 1983). Moreover, Rubin (2001) advised to select covariates associated based on previous research or findings rather than using observed outcomes in order to avoid allegations of variable selection based on estimated effects, therefore, we compiled this suite of covariates following previous work done by Joppa and Pfaff (2011), Gonzalez-Roglich et al. (2019), Feng et al. (2021) for example.

Short reference:

1. Augurzky, B. and Schmidt, C.M., 2001. The propensity score: A means to an end. Available at SSRN 270919.
2. Holmes, W.M., 2013. *Using propensity scores in quasi-experimental designs*. Sage Publications.
3. Stuart, E.A., 2010. Matching methods for causal inference: A review and a look forward. *Statistical science: a review journal of the Institute of Mathematical Statistics*, 25(1), p.1.
4. Rubin, D.B., 2001. Using propensity scores to help design observational studies: application to the tobacco litigation. *Health Services and Outcomes Research Methodology*, 2, pp.169-188.
5. Rosenbaum, P.R. and Rubin, D.B., 1983. The central role of the propensity score in observational studies for causal effects. *Biometrika*, 70(1), pp.41-55.
6. Gonzalez-Roglich, M., Zvoleff, A., Noon, M., Liniger, H., Fleiner, R., Harari, N. and Garcia, C., 2019. Synergizing global tools to monitor progress towards land degradation neutrality: Trends. *Earth and the World Overview of Conservation Approaches and Technologies sustainable land management database. Environmental science & policy*, 93, pp.34-42.
7. Feng, Y., Wang, Y., Su, H., Pan, J., Sun, Y., Zhu, J., Fang, J. and Tang, Z., 2021. Assessing the effectiveness of global protected areas based on the difference in differences model. *Ecological Indicators*, 130, p.108078.

Did you consider the effect of spatial autocorrelation on the results? If so, explain.

See

Negret, P. J. et al. Effects of spatial autocorrelation and sampling design on estimates of protected area effectiveness. *Conserv. Biol.* 34, 1452–1462. <https://doi.org/10.1111/cobi.13522> (2020). And Schleicher, J. et al. Statistical matching for conservation science. *Conserv. Biol.* 34, 538–573 549 (2020).

Thank you for this question - we appreciate the importance of addressing spatial autocorrelation in a wide number of map analyses, but considering biomass decouples very quickly with spatial scale in forests (well below 1 km) we did not account for it in this study. The matched pairs are typically very far apart (a minimum buffer of 10 km but they can be anywhere within the biome and country in question). As our study focuses on total carbon stocks within PAs compared to total carbon stocks in matched samples, and thus each group is summed, we assume that as long as spatial independence exists between each PA pixel and its matched counterfactual spatial autocorrelation will be negligible. That said, the Negret et al. reference is interesting, and of the matching approaches used in that paper, ours is closest to the independent matching (as we match by biome), which had the lowest spatial autocorrelation of the approaches explored.

We have added the following to the methods section: ‘We assume negligible spatial autocorrelation amongst PA and matched pixels, as GEDI’s AGBD product has been demonstrated to exhibit minimal spatial autocorrelation (Saarela et al., XXX) at its 1 km resolution (each 1 km pixel is comprised of several AGBD samples spaced along track at minimum 60 m). PA matches are also at minimum 10 km from counterfactuals matches. Other studies have examined the influence of spatial autocorrelation between sets of PA and counterfactual points on the perceived effectiveness of PAs to curb forest loss, and of the approaches explored our matching algorithm most closely resembles the independent matching presented by Negret et al., 2020, which was found to be the least affected by spatial autocorrelation. While a thorough analysis of potential spatial autocorrelation between counterfactuals is outside the scope of this study, considering its global scale, further attention to this matter is encouraged for national or subnational studies.’

Reviewer #3 (Remarks to the Author):

This is a significant and important paper. I believe it is the first statistically rigorous paper to assess the global impact of the world's protected area dataset on their carbon stocks, and is an innovative and important use of the GEDI spaceborne LiDAR dataset. The implications go far beyond the field of remote sensing and conservation, giving good reason for it to be published in a non-specialist journal. The figures will be important for NGOs and policy makers, and will be widely circulated.

The new evidence presented for reduced DEGRADATION rates within PAs compared to matched similar pixels in the year 2000 is important, and has not been quantitatively shown on such scale before.

Overall I find this to be a sound paper, with evidence that supports the conclusions, and no major and obvious flaws. I do have some concerns though, which I think should be fixed or clarified prior to publication.

Thank you very much for the positive feedback - please see detailed responses to your comments below.

GENERAL POINTS

GEDI does not cover the whole world, only going up to 51 degrees north/south (about the latitude of London). I do not think this is a big problem, and I do not suggest the analysis is changed. But the 'global' in the title and implied in the abstract is incorrect, as the numbers miss a number of protected areas in Canada, Russia, northern Europe (e.g. most of the UK, Scandinavia). This needs to be made clear. This could lead to the figures from the paper being used in an inaccurate way.

Thank you, this is a very important point, and we have clarified this in a few places in the paper:

- 'Global -> global scale' in the abstract
- 'GEDI collects data from the International Space Station (ISS) which covers all tropical and temperate forests, as well as the southern boreal, but does not collect data north of ~52 degrees. Thus, while the results in this paper are global-scale, they are not truly global as they omit PAs north of this latitude.' added at the end of the introduction.
- 'Additionally, North America and Europe are underestimated as PAs north of ~52 degrees latitude were not included in this study.' added to the caption of Table 1
- 'at a global scale (within the GEDI domain)' added to introduction

We considered removing the word global from the title as well, however we feel that would not distinguish the paper sufficiently from other studies conducted at local, national or even continental scales. As we feel this is the largest PA analysis of AGB to date, we would like to keep the title as it stands but caveat in the writing and figures for clarity.

Similarly Figure 2 is misleading. By showing land in northern Europe (say) along with a graph saying 'Europe', but based on data that does not extend to the northern half of Europe, the reader could be misled. I suggest cropping the world map in this figure and making it much clearer to the reader where the data are incomplete.

Thank you, we have regenerated the figure to clearly reflect the GEDI domain.

It would be useful I think for a general reader to have more understanding given in the Introduction and Results about how GEDI functions. For example pointing out its sampling

nature, and saying what proportion of each PA is actually sampled by GEDI shots, and how it varies by cloud cover and orbit patterns. A figure showing the % of the average PA in a country that is covered by GEDI shots might be useful for example? This might show more missing data in (say) Indonesia compared to less cloudy places.

Similarly, a little bit of an introduction to matching, and brief table or similar saying what datasets you used for matching, I think belongs in the main text of the paper (Introduction maybe?) not in the Methods only. Fig 6 is good, but some info on datasets could be included there, there is space.

Thank you for these suggestions! With respect to GEDI's coverage, we have taken the latest published coverage numbers from the Dubayah et al. 2020 manuscript (~70% of global cells) and added this and some more background information on the mission to the introduction:

'GEDI launched on December 5, 2018, and is collecting full waveform lidar samples from the ISS between ~52°N and 52°S under the ISS orbit (Fig 1). GEDI has three lasers operating at 1064 nm, each illuminated ~25m circular 'footprints' to produce billions of high resolution samples of surface elevation, vegetation height, and foliage distribution. GEDI is not a mapping mission, in that it does not collect data continuously over Earth's surface, but instead provides footprint samples spaced ~60 m apart along each laser track, with ~600 m spacing between tracks. Therefore not every part of every PA is mapped. 25 m samples are aggregated to 1 km estimates of AGBD (and then carbon). At the time of writing GEDI has collected sufficient data to fill ~70% of all GEDI-domain 1 km pixels (Dubayah et al. 2022).'

Regarding Table S1, we feel this is more appropriate in the methods section, and have listed the categories of matching variables in the main text. We do not want to distract readers from the main results, but we have added in-line citations to Table S1 in Figure 6 and Figure 6 notations to better direct the readers to the corresponding content.

I am not an expert in matching algorithms - from what I can see the methods seem sensible. I have two concerns though:

Was any effort made to ensure a similar number of GEDI shots per matching 1 km pixels in/out PAs?

Thank you for this question. We have attempted to ensure the same number of 1km pixels to be matched in/out PAs, but as shown in Figure S1A, the number of GEDI shots in each 1km cell varies due to the sampling nature of the GEDI mission. Although we cannot control for how many GEDI samples landed in each sample, we did set a threshold of a minimum of 5 GEDI shots/1km pixel is needed to obtain reasonable representation of the forest structure in a given cell. Moreover, this difference in number of GEDI shots would not influence our results as we

averaged the structural metrics in each 1km pixel paris in each PA for assessing PA effectiveness.

1km is a coarse resolution to deal with deforestation and degradation, as things in/out the PA could get confused. Were efforts made to ensure that every PA 1km pixel was 100% within a PA? If not, I do have worries about the whole analysis - but from my reading of the methods there is no concern here as they state that efforts were made to avoid PA boundaries so I think this was considered. And similar, I assume PAs smaller than a few 1 km pixels were excluded?

Thank you - we agree it was important to ensure analysis was strictly within PAs. in We avoided sampling the PA boundaries by implementing a 10-km buffer around any PA boundaries. This way complexities in spillover effect or displaced deforestation that typically occur around the PA boundaries are excluded for clean assessment of PA effectiveness. For PAs smaller than 1km pixels, they were filtered out when creating the 1km grid cells. Regarding the resolution of analysis, we agree that for deforestation and forest degradation studies, higher spatial resolutions are more suitable. But the purpose of this study is mainly to quantify the additional AGB and carbon stocks stored in protected forests, which is a cumulative term. The choice to stay at 1km is limited by the spatial resolution of the majority of our covariates used for matching, and by the amount of computational resources we can afford to allocate. Moreover, assessment performed at such a resolution also matched with needs of stakeholder and policy makers, and aligned with the resolution of future scientific products, such as the GEDI L4B 1 km gridded AGB (Dubayah et al., 2020).

We have added 'A 10 km buffer around any PA borders was also implemented to remove potential spillover effects e.g. encroachment at the PA boundaries or potential leakage into neighboring forests ⁶⁰⁻⁶². Therefore no locations within the 10 km buffer of any protected area were considered for matching.' to clarify this point to readers.

MINOR POINTS

Abstract talks about 'C stocks' but then gives numbers in terms of AGB. Maybe simpler for the reader to convert numbers to C in abstract?

Thank you - yes we agree and we have converted to C throughout.

First sentence in text switches between AGB and Gt CO2. Again, I would prefer a single unit throughout to avoid confusion (non-forest biomass scientists do not have the conversions between biomass, C, and CO2 ready to use in their heads as we do!) Maybe just doing everything in Mg C throughout would help?

Thank you - yes we agree and we have converted to C throughout.

I dislike the comparison in the abstract to "are roughly equivalent to annual global fossil fuel emissions". This is conflating a total stock with an annual flux. Would be better phrased as "this extra stock of carbon is roughly equivalent to a single year of global fossil fuel emissions".

Another reviewer (1) made a similar suggestion, and we completely agree. We have rephrased to note the single year.

Figure of 26% of all AGB being in protected areas is higher than 14.5% reported for the tropics in Collins & Mitchard 2017 <https://www.nature.com/articles/srep41902> Could mention this discrepancy in the paper, and explain whether due to new protected areas since then, tropics having a lower proportion protected than non-tropics, or different AGB dataset? Note there is vague mention of previous studies line 66-67, which says they fail as previous AGB datasets saturate in high biomass forest - but this is not always true, the Collins & Mitchard paper used the Avitabile et al. 2016 carbon map which does not saturate.

Thank you for this point - we feel our analysis is more exhaustive than past estimates that were based on a much sparser (and often erroneous) set of satellite observations. Additionally, the 26% refers to the entire GEDI domain rather than just the tropics, so is not directly comparable to the Collins & Mitchard estimate. We also disagree with the statement that the Avitabile e 2016 map does not saturate - maybe we are misinterpreting, but the map is essentially a combination of a few maps based on sparse ICESat GLAS samples that are then extrapolated with optical data - which is well known to saturate. In the Avitabile paper (Fig 6) the saturation is very clear. Further I do not think the authors would claim a lack of saturation for this paper, while we do not see a saturation effect in the GEDI data (orders of magnitude more lidar than GLAS provided, at a resolution designed for forests). Thus while we do not directly compare to the Collins & Mitchard estimates we think adding this comparison to the paper would be confusing as 1) we do not report for the pantropics alone and 2) we feel those estimates were based on inferior data (although best quality at the time of writing for the paper).

From Avitabile et al., 2015

Fig 1 B-E is incorrectly cropped, with an extra strip with no date covering e.g. the UK included when it is not in 1A. I think these should be cropped to match 1A

Response: Thank you very much for pointing this out, and we have modified Fig B-E to match the spatial extent of Fig 1A.

Not sure Fig 4 parcels C/D are that useful - largely a function of the area of the countries. Same things done per unit area might be useful?

Thank you - we discussed this, but feel that essentially the per unit area information is provided by the mean AGCD panel. As we report national totals we wanted to make it clear that this is often, as you point out, a simple function of larger countries hosting larger PA networks. However, this is not always the case, hence presenting both the total PA area and the mean difference in AGBC for PAs in a country.

Line 298 - REDD+ acronym wrongly spelled out: REDD+ is “Reducing emissions from deforestation and forest degradation, conservation of existing forest carbon stocks, sustainable forest management and enhancement of forest carbon stocks” according to relevant UNFCCC texts - not sure REF 55 relevant here. I don’t think this dataset necessarily helps with including protected areas under voluntary sector REDD+ projects - the fact that they’re doing better does not prove additionality on its own. But also under UNFCCC REDD+ there’s no reason they can’t be in anyway (“conservation of existing carbon stocks”). Suggest rewrite these couple of sentences.

We agree, this gets a bit murky so for clarity we have removed that sentence to have a clearer link to the latest agreements.

Reviewer comments, further round review

Reviewer #1 (Remarks to the Author):

I greatly appreciate the authors thorough response to my previous concerns and suggestions. I am pleased with the resulting draft and have no outstanding concerns.

Reviewer #2 (Remarks to the Author):

The authors have made substantial efforts to revise the manuscript based on the comments raised in the first round of review. The responses to the reviewers' comments were clearly addressed on all occasions, so a solid effort was spent on revising.

General: There are a lot of acronyms used throughout. While these may be necessary, the use of two acronyms consecutively is hard to digest and I suggest rewording 'PA C'

ISS – The name is written in full after the acronym appears earlier

Abstract: Consider using small caps for 'protected areas' in the abstract?

Intro

Lines 40-44. Consider rewording sentence (especially the beginning part which lists the frameworks and policies without anchoring in context).

Discussion

Lines 311-313: While I agree there are multiple benefits possible from establishing PAs, this paper is only exploring carbon benefits, and links to biodiversity benefits in the literature. This final statement reads to me as a call to action to expand PAs, regardless of where they are placed, how well they are managed, and whether communities benefit from them or not. A little more clarity in this final statement would strengthen the conclusion.

REVIEWERS' COMMENTS

Reviewer #1 (Remarks to the Author):

I greatly appreciate the authors thorough response to my previous concerns and suggestions. I am pleased with the resulting draft and have no outstanding concerns.

Many thanks!

Reviewer #2 (Remarks to the Author):

The authors have made substantial efforts to revise the manuscript based on the comments raised in the first round of review. The responses to the reviewers' comments were clearly addressed on all occasions, so a solid effort was spent on revising.

General: There are a lot of acronyms used throughout. While these may be necessary, the use of two acronyms consecutively is hard to digest and I suggest rewording 'PA C'

Thank you – we have reworded as PA aboveground C (following an editorial note that we do not include soil C)

ISS – The name is written in full after the acronym appears earlier
Done.

Abstract: Consider using small caps for 'protected areas' in the abstract?
We have made this change.

Intro

Lines 40-44. Consider rewording sentence (especially the beginning part which lists the frameworks and policies without anchoring in context).

We have reworded as 'Several policy frameworks emphasize that habitat conservation and restoration should contribute simultaneously to biodiversity conservation and climate change mitigation^a. These frameworks include the UN Sustainable Development Goals (SDGs), decisions under the United Nations Framework Convention on Climate Change (UNFCCC) and the Convention on Biological Diversity (CBD).'

Discussion

Lines 311-313: While I agree there are multiple benefits possible from establishing PAs, this paper is only exploring carbon benefits, and links to biodiversity benefits in the literature. This final statement reads to me as a call to action to expand PAs, regardless of where they are placed, how well they are managed, and whether communities benefit from them or not. A little more clarity in this final statement would strengthen the conclusion.

We agree that carbon benefits are the primary focus of this study, but we also analyzed forest height, canopy cover and plant area index, which are related to habitat and therefore biodiversity. Rather than edit the last sentence, we have added the following to the final paragraph to clarify the linkages to non-carbon benefits: 'We also demonstrate that other forest structure metrics related to habitat (forest height, canopy cover, and PAI) are also preserved by PAs, suggesting effective co-benefits of PAs include both climate and biodiversity.'